# Interventions to improve primary healthcare in rural settings: A scoping review

**Kris Aubrey-Basler**[1,2,3‡]*, **Krystal Bursey**[2‡], **Andrea Pike**[1,2], **Carla Penney**[2],
**Bradley Furlong**[2], **Mark Howells**[2], **Harith Al-Obaid**[2], **James Rourke**[1,2],
**Shabnam Asghari**[1,2,3], **Amanda Hall**[1,2,3]

**1** Discipline of Family Medicine, Faculty of Medicine, Memorial University, St. John's, Newfoundland and Labrador, Canada, **2** Primary Healthcare Research Unit, Faculty of Medicine, Memorial University, St. John's, Newfoundland and Labrador, Canada, **3** Division of Public Health and Applied Health Sciences, Faculty of Medicine, Memorial University, St. John's, Newfoundland and Labrador, Canada

‡ KAB and KB are contributed equally to this work as co-first authors.
* kaubrey@mun.ca

## Abstract

### Background

Residents of rural areas have poorer health status, less healthy behaviours and higher mortality than urban dwellers, issues which are commonly addressed in primary care. Strengthening primary care may be an important tool to improve the health status of rural populations.

### Objective

Synthesize and categorize studies that examine interventions to improve rural primary care.

### Eligibility criteria

Experimental or observational studies published between January 1, 1996 and December 2022 that include an historical or concurrent control comparison.

### Sources of evidence

Pubmed, CINAHL, Cochrane Library, Embase.

### Charting methods

We extracted and charted data by broad category (quality, access and efficiency), study design, country of origin, publication year, aim, health condition and type of intervention studied. We assigned multiple categories to a study where relevant.

### Results

372 papers met our inclusion criteria, divided among quality (82%), access (20%) and efficiency (13%) categories. A majority of papers were completed in the USA (40%), Australia (15%), China (7%) or Canada (6%). 35 (9%) papers came from countries in Africa. The most common study design was an uncontrolled before-and-after comparison (32%) and

**Data Availability Statement:** The data are all contained within the manuscript and its Supporting information files.

**Funding:** This study was funded by the Canadian Institutes of Health Research (grant # 141676) awarded to KAB. The funders (https://cihr-irsc.gc.ca/e/193.html) had no role in study design, data collection and analysis, decision to publish, or preparation of the manuscript.

**Competing interests:** KAB and JR are family physicians who previously practiced in rural areas. KAB, JR, and SA are rural health services researchers with an interest in the equity of health service distribution. The authors have no other disclosures to report.

only 24% of studies used randomized designs. The number of publications each year has increased markedly over the study period from 1-2/year in 1997–99 to a peak of 49 papers in 2017.

## Conclusions

Despite substantial inequity in health outcomes associated with rural living, very little attention is paid to rural primary care in the scientific literature. Very few studies of rural primary care use randomized designs.

## Introduction

The United Nations estimates that about 3.4 billion people worldwide live in rural areas [1]. Residents of these areas have poorer health status, are more likely to live in poor socio-economic conditions, demonstrate less healthy behaviours, and tend to have higher mortality rates than people who live in urban centres [1–6]. These disparities in health outcomes indicate a greater need for health services that address these factors, and these services are usually delivered under the umbrella of primary care [7]. Countries and regions with stronger primary care tend to have improved health outcomes, lower health system costs and reduced inequity in health [7, 8]. It is important to note that the term primary care has many definitions and is sometimes used interchangeably with the term primary healthcare. For the purposes of this scoping review, we aimed to adopt a broad definition to capture a wide range of primary care services. We used the Canadian Foundation for Healthcare Improvement (CFHI) policy document (Toward a Primary Care Strategy for Canada) to guide many of the definitions used in this review [9]. As such we adopted CFHI's definition of primary care: "an inclusive term to cover the spectrum of first-contact healthcare models from those whose focus is comprehensive, person-centered care, sustained over time, to those that also incorporate health promotion, community development and intersectoral action to address the social determinants of health" to serve as the working definition for this scoping review [9].

Despite increased needs, rural populations are not as well-served as their urban counterparts and they experience more difficulty accessing primary care [10]. Lack of access to primary care in rural areas is in part due to the difficulty of retaining physicians and other healthcare professionals in their communities [11]. Additionally, residents living in rural communities often have to travel in order to access healthcare. This can pose an added burden to individuals who do not have reliable transportation or have mobility issues. When travel is required to access services, it may mean people have to leave their communities and families and incur additional costs for accommodation and meals. Even when a person can access primary healthcare in their community, they may encounter additional barriers such as difficulty contacting the physician or clinic to schedule an appointment and long wait times for appointments [12]. Specialist care is even less equitably distributed than primary care, suggesting a greater need for services, such as an expanded scope of practice for rural primary care providers that can replace some of the care usually offered by specialists. The World Health Organization highlights that remote populations face significant health disparities compared to both urban and many rural areas due to their smaller size, isolation, and socioeconomic disadvantages [13]. Geographic isolation and sociocultural differences also intensify healthcare shortages in remote areas. Thus, addressing these challenges requires interventions focused on increasing healthcare access, ensuring equitable quality of care, and prioritizing patient-centered approaches.

Improving primary care is an important step toward improving health outcomes in rural and remote communities, and this will likely require a multifaceted approach. This scoping review synthesizes and categorizes studies that have evaluated interventions to improve primary care in rural and remote settings and it is the first comprehensive review to do so. Because of the breadth of literature we expected to find in terms of number of papers and diversity of study designs, interventions and outcomes, we elected to use scoping methodology with high-level data extraction rather than more detailed extraction and synthesis of a systematic review [14]. Following guidance for scoping reviews as described in our methods below, the purpose of this review is to provide an overview of the types of available evidence regarding interventions to improve primary care in rural and remote settings and identify the key characteristics of this body of literature (e.g., the health topics and interventions studied), health topic areas ready for systematic review, and knowledge gaps.

## Method

### Design

We used the refined Arksey and O'Malley [15] six-stage framework to structure our methods [16]. We followed the Preferred Reporting Items for Systematic reviews and Meta-Analyses extension for Scoping Reviews (PRISMA-ScR) checklist and the Johanna Briggs Institute manual to enhance review quality and reporting [14, 17].

**Stage 1: Developing the research question.**   We were interested in describing all studies that reported on the evaluation of a program or intervention designed to improve health care and that was implemented in a rural or remote primary care setting. We organized the papers by the outcomes studied, according to three broad categories listed by a prominent primary care policy document [9]. We assigned papers to multiple categories where appropriate:

1. Quality of primary care: We classified studies addressing healthcare quality loosely based on the Institute of Medicine's "domains of health care quality [18]." The efficiency domain was further divided and included in a separate heading below.

    a. Evidence-based practice

    b. Clinical outcomes

    c. Patient experience

    d. Equity of care

    e. Patient safety

2. Access to primary care: Multiple policy documents recommend that patients have access to a regular, identifiable primary care provider or team and timely access to primary care through arrangements that facilitate 24/7 access to appropriate services [9, 13, 19]. We thus subdivided this statement into the following two access topics:

    a. Access to a primary care provider

    b. Access to primary care services

3. Efficiency of primary care: The primary care system should continually seek to reduce waste and cost of supplies, equipment, space, capital, ideas, time and opportunities [9]. This broad category includes the following outcome variables:

    a. Wait Times

b. Healthcare utilization

c. Costing and cost-effectiveness

*Setting.* There is broad disagreement among researchers, policy makers, and community members about the best definition of rural, and the appropriate definition likely varies with the circumstances [20]. To maximize inclusiveness for this review, we have elected to accept any program described as rural or remote by the paper's authors. As described in the introduction we used the CFHI's definition of primary care, "an inclusive term to cover the spectrum of first-contact healthcare models from those whose focus is comprehensive, person-centered care, sustained over time, to those that also incorporate health promotion, community development and intersectoral action to address the social determinants of health" as the working definition for this review [9]. This definition allows for a broad range of interventions for inclusion; examples of interventions that were included and excluded following this definition can be found in Table 1. Additionally, in piloting this definition during our screening process, we realized reporting was not always clear and we amended the definition to maximise study inclusion to also accept any program described as being implemented in primary care by the paper's authors.

*Study design and language.* The search was limited to papers written in English. All randomised and non-randomised experimental or observational designs were included (e.g., randomised controlled trials, controlled before-after studies, uncontrolled before-after studies, interrupted time series and observational/cohort studies). Non-experimental studies such as descriptive studies or chart reviews were excluded. We developed a protocol a priori but did not publish prospectively.

**Table 1. Screening template for full text study review.**

| Inclusion criteria | Exclusion criteria |
|---|---|
| Population: Any patients | Population: No exclusions |
| Setting: Primary care–if the intervention is delivered:<br>• by a family physician (either individually or as part of a comprehensive/collaborative care team) or by another primary care provider or<br>• was delivered as part of a public health community intervention such as screening but involved a primary care service at some part of the delivery or<br>• in a setting that is described as primary care in the individual study | Setting: Primary care–if the intervention is delivered:<br>• within normal hospital services, emergency medicine, paramedicine, dental care<br>Note: exceptions are if the primary care service/provider is situated within a hospital |
| Setting: Rural/remote geographic areas<br>• as defined in each paper | Setting: Rural/remote geographic areas<br>• urban geographic area |
| Study designs: All randomised and non-randomised experimental or observational study designs<br>• randomised controlled trials<br>• controlled before-after studies<br>• uncontrolled before-after studies<br>• interrupted time series<br>• observational/cohort studies | Study designs: All non-experimental studies (those without a comparison group)<br>• descriptive studies<br>• chart reviews |
| Outcomes:<br>• Access to primary care<br>• Quality of care<br>• Efficiency/cost of care | Outcomes: Any outcomes unrelated to access, quality or efficiency. |

**Stage 2: Identifying relevant studies.** An experienced information specialist completed an electronic literature search for articles reporting the evaluation of a program or intervention to improve rural healthcare, initially between January 1, 1996 and December 30, 2022 and cataloged in the following databases: Cumulative Index to Nursing and Allied Health Literature (CINAHL), the Cochrane Library, Embase, and PubMed. A sample electronic search strategy is included in S1 Appendix.

**Stage 3: Study selection.** Articles were downloaded to Covidence systematic review software [21] and duplicates removed. Initial screening of titles and abstracts (and full-text review) was completed by two reviewers independently using pre-specified eligibility criteria (Table 1). During both these screening stages, a third reviewer mediated disagreements to reach a final decision about study inclusion.

**Stage 4: Charting the data.** Data extraction from each paper was initially completed by one reviewer (MH, HO, BF, CP) and double checked by a second author (KB, AP, BF) using a data extraction template. Discrepancies were reviewed with a senior author (AH). Data related to study characteristics, publication year, country, study design, study aim, intervention description, and types of outcomes assessed including assessment methods and measurement tools was extracted verbatim. The information was coded where possible; the intervention description was assigned a category code that was defined by the author team after reviewing all interventions descriptions for overarching themes as described in Table 2, and the outcomes were coded into one of the twelve possible outcomes within the three major categories as defined on page 4 under the subsection "Stage 1: Developing the Research Question".

**Stage 5: Collating, summarizing and reporting the results.** An overall summary of all included studies was conducted using descriptive statistics to summarize study characteristics, evaluation characteristics and types of outcome categories assessed. For each of the three outcome categories (i.e., access, quality and efficiency), studies were summarised together according to the sub-outcome measured within that category. For each sub-outcome, a tabular summary of the individual study characteristics, aim, intervention code and description and the outcome assessment information were provided. Where applicable, the tabular summaries are organised by the health topic studied. Due to the large amount of data, we have included these tabular summaries in the online supplementary files. Consistent with guidance on scoping reviews, we did not appraise methodological quality or risk of bias in the included articles [14, 15].

In addition to summarizing study characteristics and health topic areas covered within each of the main outcomes, we also completed a content analysis for each outcome category to provide readers with a high-level content summary, organized by health topic. For this analysis, we included health topics that were assessed in three or more studies. We also assessed this literature for its readiness for systematic or in-depth scoping review and to identify knowledge gaps—those areas where additional research is required. The criteria we used to make these judgements is presented in Box 1.

**Stage 6: Stakeholder engagement.** This review is a product of consultation between one of our authors (JR) and the Atlantic Canadian premiers and was completed with guidance from the Atlantic Canadian Ministries of Health. In an effort to progress on evidence-informed rural health policy, these stakeholders were interested to know the overall state of science regarding rural primary care. After compiling the results of the review, members of our team met with additional health policy and programming stakeholders to review the broad state of rural primary healthcare knowledge. In so doing, they also helped us determine areas within the broader body of literature where a deeper, narrative review of the literature would be helpful, which our team will focus on in several future planned reviews.

**Table 2. Definition of intervention codes.**

| Intervention Code | Definition |
|---|---|
| Medical education (selective recruitment) | A medical education program that is tailored to training rural physicians OR has selective criteria for accepting rural students OR offers a program tailored to recruit and retain rural physicians |
| Medical education (exposure to rural practice) | A medical education program for training in family medicine or that integrates rural curriculum content with the opportunity to train in a rural setting |
| Financial incentive | The use of scholarships or "loan-forgiveness" for prospective medical students as a method of recruitment to a particular medical education program OR offering a financial incentive to physicians to encourage them to practice and remain in a rural area OR using a financial incentive to promote adherence to evidence-based guidelines. |
| Well-being | The introduction or development of a program to support the social and psychological well-being of practicing rural physicians as a method to encourage them to continue practicing in rural areas (including improving working conditions). |
| Expanded scope of practice | Health care providers taking on a new skill or provide a new service (e.g., nurse taking on a responsibility normally covered by a family doctor or a family doctor providing more specialized care). |
| Telehealth or virtual care | Using telehealth or other virtual systems to administer primary care appointments OR using technology to allow patients to access their health information or book/have appointments with healthcare professionals online. |
| Coordination/referral pathways | Increase referral to existing services by improving awareness or easing the referral process, etc. |
| Training of lay-persons | Recruiting and training members of the community to provide a set of health care and/or health care coordination services for patients in the community OR to implement best practices for a particular health topic/condition. |
| Healthcare provider training | Healthcare providers completing additional education on evidence-based practices for a particular health topic. |
| Transportation | Implementing a new transportation service to improve access or care for rural patients. |
| Decision support | Intervention to assist healthcare provider decision making and/or inform healthcare providers of best practices regarding a healthcare topic or condition. |
| Reorganization of services | Reorganization of healthcare services to better align with evidence-based guidelines or to use a previous service or treatment in a novel way to optimize care. |
| Increasing staff resources | Increase of staff allocation or improved clinic resources to improve a service. |
| Patient education/navigation | Patients completing additional education on evidence-based practices for a particular health topic or condition relating to their own health. This could include group or individual sessions with a knowledgeable healthcare provider, or the distribution of educational materials. |
| Screening | Increased or added screening measures in order to adhere to guidelines |
| Audit and feedback | Analyzing a targeted behaviour/performance over a specific period and providing feedback on the behaviour/performance to health professionals. |
| Implementing a new service | Implementing a novel service to address a healthcare need or improve upon previous care. |

**Note:** Intervention codes were defined by the review team. They were based on similarities among interventions included in the papers and refined to produce the best overall descriptor code and definition.

## Results

The literature search returned 6,592 citations; 6,220 of those were excluded (Fig 1) leaving 372 articles for data extraction and synthesis [22–390]. Details on article characteristics are presented in S2 Appendix. Briefly, the majority of articles were published after 2010 with an

**Box 1: Judgement criteria to assess directions for future research.**

| | |
|---|---|
| Systematic review | We recommended systematic review when we found at least three studies on a health topic with apparent similarity in interventions and outcomes that could be used to form a clear PICO question. |
| In-depth scoping review | We recommended in-depth scoping review for health topics for which clear PICO questions cannot be easily discerned (i.e., there is high variation across outcomes and interventions across the identified studies). |
| Additional research required (knowledge gaps) | We recommended additional research for topic areas in which: 1) higher quality (controlled) studies are feasible but not evident in the literature base 2) controlled studies may not be feasible, but for which the quality of uncontrolled studies could be improved 3) there were few studies all of which were investigating different interventions and outcomes 4) there were very few studies evident in the literature base. |

increase in the number per year beginning in 2015 that peaked in 2017 before beginning to decline (Fig 2). The majority were conducted in the United States (40%), Australia (15%) and China (7%) (Table 3). Many study designs were used to evaluate interventions, the most common of which were uncontrolled before-and-after (32%) followed by randomized trials (24%), prospective or retrospective cohort (18%), controlled before-and-after (11%), and cross-sectional studies (11%). In terms of outcome categories, the majority of studies (n = 306) assessed outcomes relating to quality of care focusing on improving desired health outcomes for patients and adherence to evidence-based standards (S3 Appendix). Seventy-six studies assessed access to primary care largely focusing on recruitment and retention of family physicians and access to services (S4 Appendix). Forty-seven studies assessed outcomes related to improving the efficiency of primary care services in rural areas (e.g., cost of delivery, healthcare use; S5 Appendix). A description of the studies for each outcome category is presented in the following sections.

## Quality of care

Three-hundred and six articles (82%) reported on interventions aimed at improving the quality of primary care provided in a rural/remote setting. Seventy-two percent (n = 230) focused on improving the health outcomes of patients; 30% (n = 112) aimed to improve adherence to or adoption of evidence-based practices, 11% (n = 34) aimed to improve quality of care via enhancing patient experience, 1% (n = 4) studies aimed to improve equity of care, and only one study assessed patient safety. A detailed breakdown of all intervention types, study designs, individual study outcomes, and outcome measurements for quality-of-care studies can be found in S3–S7 Appendices. We have summarized the data in the following sections.

**Evidence-based practice (n = 112).** Nearly three quarters of the 112 articles that aimed to improve quality of care via improving adherence to/adoption of evidence-based practices were published between 2010–2022. They were conducted in a broad range of countries, but just over half were conducted in either the United States (n = 35) or Australia (n = 20). Provider education/training (n = 35), improving coordination/referral pathways (n = 14), and patient education (n = 12) were the most frequent interventions employed. The studies were largely uncontrolled before-after studies (n = 42), randomized controlled trials (n = 32). The most commonly studied guidelines or recommendations were related to diabetes (n = 21), mental health care (n = 13), cancer screening (n = 12). Table 4 provides a content summary analysis for the nine health topics that were assessed in three or more studies. Of note, several papers

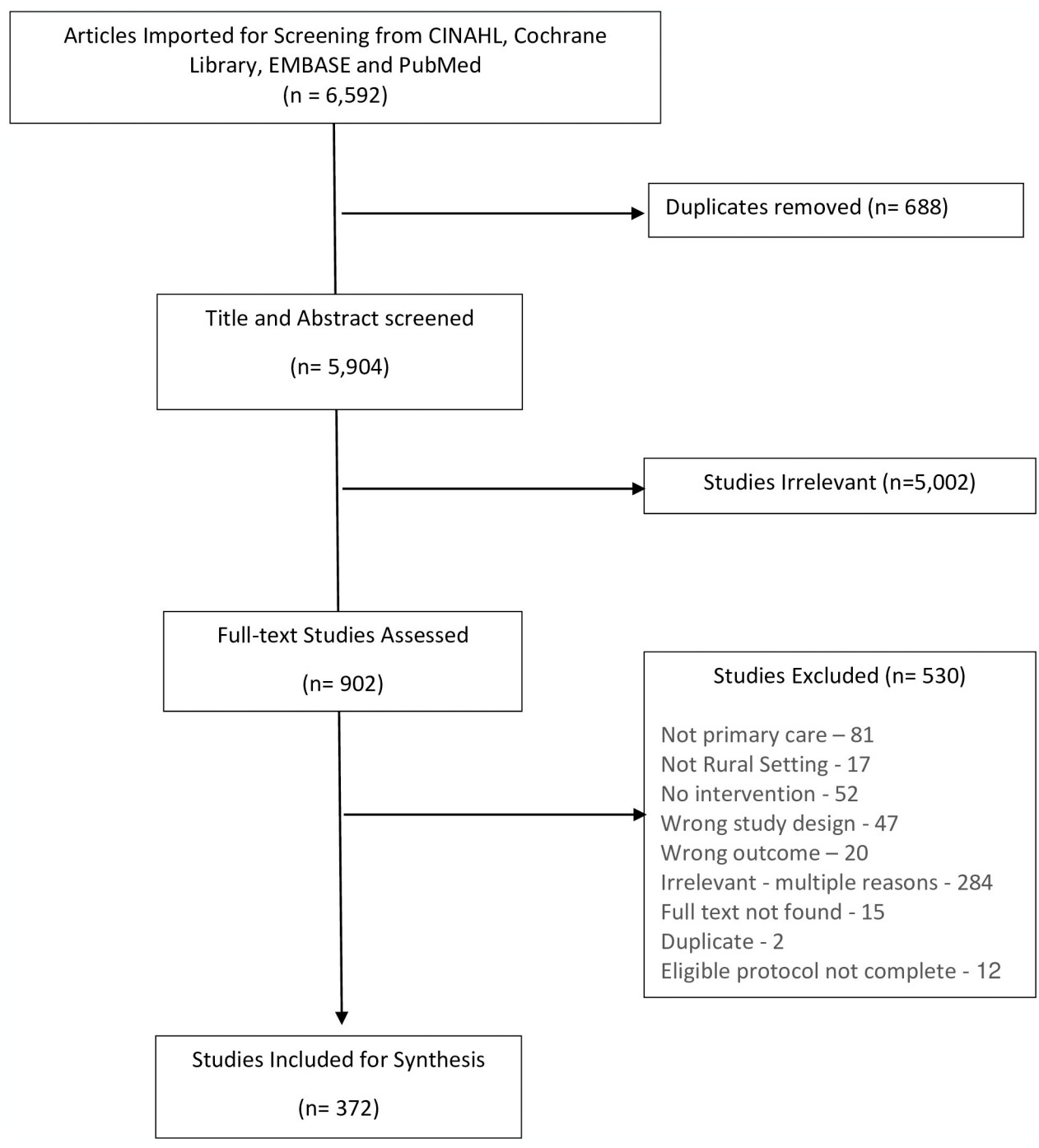

**Fig 1. Prisma flow diagram depicting flow of studies through the screening process.**

did not assess a specific health topic (e.g., they were studying an intervention applicable to any condition). Information on these studies can be found in S3 Appendix. We judged three topics to be ready for systematic review: cancer (n = 12), CVD (n = 5) and respiratory infections (n = 4). Meta-analysis for each of these topics may not be possible and they would therefore

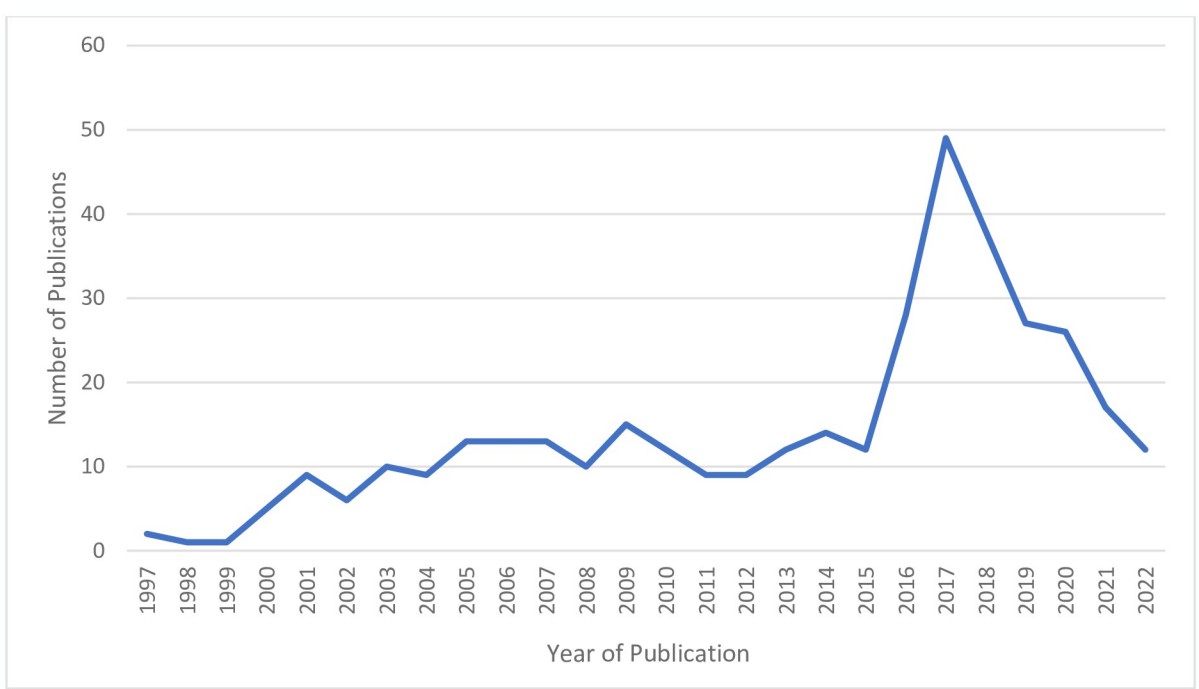

**Fig 2. Number of publications by year.**

require some degree of narrative synthesis based on differences in interventions. We judged an additional 4 topic areas that would benefit from in-depth scoping review including diabetes (n = 21), mental health (n = 13), asthma (n = 6) and pregnancy-related studies (n = 4). We believe additional research is required for chronic disease (n = 3) and HIV (n = 3). For both these topic areas the studies we found in both areas were not testing similar interventions or assessing similar outcomes. Additionally, we found very few studies in 23 topic areas including antibiotic prescribing, atrial fibrillation, back pain, chronic lung disease, COPD, dental care, epilepsy, hypertension, malaria, maternal and child health, neonatal care, obstetrics, palliative care, PCOS, post-abortion care, preventative care, sexually-transmitted infections, skin infections, sleep, speech impairments, surgery, vaccinations, vision impairments.

**Clinical outcomes (n = 230).** Just over two thirds of the 158 studies focused on improving quality of care via improvements to clinical outcomes were published between 2010 and 2022. The studies were conducted in 45 different countries, with the majority originating in the United States (n = 100), Australia (n = 30), China (n = 16) and Canada (n = 11). Interventions commonly involved educating patients on evidence-based practices for a particular health topic or condition relating to their own health (n = 43), reorganizing health services to optimize care (n = 29), expanding the scope of practice for healthcare professionals (n = 24), upskilling or training healthcare professionals on evidence-based practices for a particular topic (n = 22), implementing a new service (n = 14), and improving co-ordination/referral pathways (n = 14). Some papers examined a single intervention while others examined a combined intervention (e.g., provider education/training + patient education), and papers may therefore be counted multiple times in the list above. The most common conditions included diabetes (n = 48), mental health (n = 34), hypertension (n = 13), and cardiovascular disease (n = 12). The most common study designs included uncontrolled before-after (n = 75) and randomized controlled trials (n = 68). Other designs reported included controlled before-after studies

**Table 3. Number of studies by economic status, country, and geographic region.**

| Economic status | East Asia and Pacific (N = 95) | Europe and Central Asia (N = 34) | Latin America and the Caribbean (N = 10) | Middle East and North Africa (N = 11) | North America (N = 172) | South Asia (N = 14) | Sub-Saharan Africa (N = 40) |
|---|---|---|---|---|---|---|---|
| Low (N = 15) | | | | | | | Burkina Faso (N = 5)<br>Ethiopia (N = 4)<br>Mali (N = 1)<br>Niger (N = 1)<br>Rwanda (N = 2)<br>Uganda (N = 2) |
| Lower-Middle (N = 33) | Philippines (N = 1) | | Bolivia (N = 1)<br>Haiti (N = 1) | Egypt (N = 1)<br>Iran (N = 1)<br>Lebanon (N = 1) | | India (N = 11)<br>Nepal (N = 2)<br>Pakistan (N = 1) | Cameroon (N = 1)<br>Eswatini (N = 1)<br>Ghana (N = 1)<br>Kenya (N = 3)<br>Lesotho (N = 1)<br>Nigeria (N = 4)<br>Tanzania (N = 1)<br>Zambia (N = 1) |
| Upper-Middle (N = 47) | China (N = 26)<br>Indonesia (N = 1)<br>Thailand (N = 1) | | Brazil (N = 1)<br>Argentina (N = 1)<br>Costa Rica (N = 1)<br>Dominican Republic (N = 1)<br>Guatemala (N = 1)<br>Mexico (N = 2) | | | | South Africa (N = 12) |
| High (N = 276) | Australia (N = 54)<br>Japan (N = 3)<br>Korea (N = 2)<br>New Zealand (N = 5)<br>Taiwan (N = 2) | Croatia (N = 1)<br>France (N = 3)<br>Germany (N = 3)<br>Greece (N = 2)<br>Ireland (N = 2)<br>Norway (N = 4)<br>Poland (N = 1)<br>Spain (N = 3)<br>Sweden (N = 2)<br>United Kingdom (N = 13) | Chile (N = 1) | Israel (N = 2)<br>Saudi Arabia (N = 1) | Canada (N = 22)<br>United States (N = 150) | | |

Notes: The total N in this table will not add up to the total N for the paper as some studies were conducted in multiple countries. Economic status was determined using the World Bank's classification system (see here for more details: https://datahelpdesk.worldbank.org/knowledgebase/topics/19280-country-classificatio

(n = 24), cross-sectional studies (n = 16), prospective cohort studies (n = 15), cohort studies (n = 14), retrospective cohort studies (n = 11), interrupted time series (n = 5) and Markov modelling (n = 2). Table 5 provides a content summary analysis for the 17 health topics that were assessed in 3 or more studies. Of note, several papers did not assess a specific health topic (e.g., they were studying an intervention applicable to any condition). Information on these studies can be found in S4 Appendix. We judged 7 topic areas to be ready for formal systematic review including CVD (n = 12), weight management (n = 9), pain management (n = 8), palliative care (n = 3), alcohol use (n = 3), stroke rehabilitation (n = 3), vaccination (n = 4), and sexually transmitted infections (n = 3). We judged 4 topic areas that would clearly benefit from an in-depth scoping review including diabetes (n = 48), mental health (n = 34), hypertension (n = 13) and cancer (n = 8). Additional research is required in four topic areas: HIV (n = 7), asthma (n = 6), chronic disease (n = 4), pregnancy (n = 3) and COPD (n = 3). Within each topic area there was a need for studies using controlled designs. The interventions and outcomes reported within each area were also highly variable. Additionally, we found very few studies in numerous areas including atrial fibrillation, burn rehab, chagas disease, childhood

**Table 4. Content summary for quality—Evidence-based medicine (health topics that were assessed by 3 or more studies are reported in the content summary).**

| Topic Study (n) | Countries, Year Range | Intervention | Study Design | Adherence* measured? Y/N/ U & Type | Content Summary | Suggestions for future research |
|---|---|---|---|---|---|---|
| Diabetes (n = 21) 2000–2021 | US (8), China (5), Aus (3), Cameroon (1), CAN (1), NZ (1), Philippines (1), Spain (1) | Service Changes (10) Provider training +/- other strategies (8) Patient education (3) | RCT (7) CS (3) US (11) | Yes (6) Unclear (15) Varied | Twenty-one studies assessed interventions that aimed to improve adherence to some aspect of guideline-based care for diabetes. There was wide variation in or lack of reporting for the aspect of care that was trying to be improved. The interventions also varied but provider education was the most consistent. Service changes was also popular but there was wide variation in the types of changes used. 30% used a RCT design and a further 14% used a control group but over half were uncontrolled. | **In-depth scoping review** There are a number of provider training interventions that have been tested using an RCT. However, there is a degree of variation in the interventions tested in this category that would benefit from further investigation to form meaningful PICOs for a systematic review. |
| Mental Health (n = 13) | Aus (4), US (4), India (2), Canada (1), Mali (1), South Africa (1) | Provider training +/- other strategies (8) Service Changes (5) | RCT (2) CS (1) US (10) | Yes (7) Unclear (2) Varied Other-Provider knowledge/ attitudes/skills (6) | Thirteen studies assessed interventions that aimed to improve adherence to some aspect to guideline-based care for mental health. Just over half used provider training as the intervention. The remaining intervention types included some change in service structure but were assessed in less than 3 studies. For the provider training interventions 2 used an RCT design and 1 used a controlled before-after study and the remainder were uncontrolled studies. | **In-depth scoping review** There are a number of provider training or service changes interventions evaluated, but most in uncontrolled studies using a range of different outcomes. A scoping review would be beneficial to form meaningful PICOs for future studies or a systematic review. |
| Cancer (n = 12) 1999–2017 | US (8), Egypt (1), Greece (1), NZ (1), UK (1) | Service Changes (5) Provider training +/- other strategies (4) Patient education (3) | RCT (5) CS (1) US (6) | Yes (10) Unclear (2) Screening rates or referral (10) | Twelve studies assessed interventions that aimed to improve adherence to some aspect to guideline-based care for cancer. The main guideline was screening rates or referral to a screening procedure such as colonoscopy. Just over half used provider or patient education and the remainder used some change to service structure or mass screening or audit and feedback but were each assessed in less than 3 studies. | **Systematic review** Four provider training interventions, all in the US and all RCTs, show promise for a systematic review. However, aims and outcomes will need more assessment for a meta-analysis. Other interventions would be suited to a narrative synthesis. |
| Asthma (n = 6) 2001–2017 | US (3), CAN (2), Aus (1) | Provider training +/- other strategies (4) Patient education (2) | RCT (1) CS (1) US (4) | Yes (4) Unclear (2) Varied | Six studies assessed interventions that aimed to improve adherence to some aspect to guideline-based care for asthma. 4 studies used provider education and 2 used patient education interventions. Multiple adherence outcomes were assessed, however, completion of Asthma action plans used in at least 3 studies. There was only one study that used a RCT design and one that used a control group the remainder were uncontrolled. | **In-depth scoping review** |

*(Continued)*

**Table 4.** (Continued)

| Topic Study (n) | Countries, Year Range | Intervention | Study Design | Adherence* measured? Y/N/ U & Type | Content Summary | Suggestions for future research |
|---|---|---|---|---|---|---|
| CVD (n = 5) 2011–2019 | China (2), Aus (1), Indonesia (1), NZ (1) | Service Changes (3) Patient education (2) | RCT (2) CS (2) US (1) | Yes (5) Varied | Five studies assessed interventions that aimed to improve adherence to some aspect to guideline-based care for CVD. Four studies had a control group, 2 of which were RCTs. No intervention type was assessed in 3 or more studies. | **Systematic review** Due to heterogeneity of interventions, a narrative synthesis may be warranted. |
| Pregnancy related (n = 4) 2016–2018 | Burkina Faso (1), Dominican Republic (1), Ethiopia (1), Guatemala (1) | Service changes (4) | RCT (1) US (3) | Yes (3) Unclear (1) Varied | Four studies assessed interventions that aimed to improve adherence to some aspect to guideline-based care for pregnancy related topics. One study was an RCT and the rest were uncontrolled. No intervention type was assessed in 3 or more studies | **Additional research** |
| Respiratory infection (n = 4) 2017–2019 | China (3), US (1) | Provider training +/- other strategies (3) Patient education (1) | RCT (3) US (1) | Yes (3) Unclear (1) Antibiotic prescribing (3) | Four studies assessed interventions that aimed to improve adherence to some aspect to guideline-based care for respiratory tract infections. three studies were RCTs and one was uncontrolled. Provider education with and without patient education was assessed in 3 studies. | **Systematic review** |
| Chronic Disease (n = 3) 2009–2017 | Aus (2), CAN (1) | Service Change (3) | RCT (1) US (2) | Yes (3) Varied | Three studies assessed interventions that aimed to improve adherence to some aspect to guideline-based care for chronic disease. One study was an RCT and two were uncontrolled. No intervention type was assessed in 3 or more studies. | **Additional research** |
| HIV (n = 3) 2016–2020 | South Africa (3) | Provider training +/- other strategies (1) Patient education (1) Service Changes (1) | RCT (1) US (2) | Yes (3) Varied | Three studies assessed interventions that aimed to improve adherence to some aspect to guideline-based care for HIV. One study was an RCT and two were uncontrolled. No intervention type was assessed in 3 or more studies. | **Additional research** |

**Notes**:

*Guideline adherence*: This outcome would be a measure of provider adherence to the particular guideline, recommendation, medical directive or policy change that the intervention was trying to change. It may be reported as adherence to individual recommendation(s) or a composite score of multiple recommendations. The recommendations may not be the same across studies and this does not preclude meta-analysis depending on the question of interest.

*Study design*: RCTs include individual and cluster RCTs, CS include non-randomized studies with a control group (i.e. controlled before-after studies and interrupted time series, US includes all studies without a pre-assigned control group (uncontrolled before-after, cohort studies or cross sectional studies)

*Interventions*: Due to the many types of interventions that could not all be easily listed in this table, "service change" was used to capture any of the intervention codes which involved changing some aspect of the health system (extended scope of practice, telehealth/virtual care, co-ordination/referral pathways, reorganization of service or implementation of a new service).

*Data not included in this table*: *Health topics that were assessed in 2 or less studies include*: Antibiotic prescribing, Atrial fibrillation, Back pain, chronic lung disease, COPD, Dental care, epilepsy, Hypertension, malaria, maternal and child health, neonatal care, obstetrics, palliative care, PCOS, post abortion care, preventative care, sexually transmitted infections, skin infections, sleep, speech impairments, surgery, vaccinations, vision impairments. *Topic areas without a specific health topic*: Three health areas did not include a specific health topic and were not included for this content summary: general health without a clear health topic and medication related topics.

**Table 5. Content summary quality—Patient clinical outcomes (health topics that were assessed by 3 or more studies are reported in the content summary).**

| Topic Study (n) Year Range | Countries, Year Range | Intervention | Study Designs | Outcome | Content Summary | Suggestions for future research |
|---|---|---|---|---|---|---|
| Diabetes (n = 48) 2001–2022 | US (21), Aus (9), China (4), Can (3), Spain (2), Brazil (1), Cameroon (1), Costa Rica (1), Israel (1), NZ (1), Pakistan (1), Philippines (1), Saudi Arabia (1) | Service change (23) Patient education +/- other strategies (21) Provider training (4) | RCT (13) CS (6) US (29) | HBA1C (30) | 48 studies assessed interventions that aimed to improve at least one patient clinical outcome associated with diabetes (e.g. HBA1c). Other common outcomes included fasting plasma glucose, weight, height, cholesterol and blood pressure. Patient education (with or without additional strategies) was the most frequently used intervention. Interventions related to service change in some way including extending scope of practice a non-physician to deliver a diabetes primary care service and lastly provider training (with or without another strategy) was also used. 27% of studies used a RCT design with a further 13% using a control group. The remainder were uncontrolled designs. | **In-depth scoping review.** This area shows promise for a *systematic review and potential meta-analysis*. However due to the wide variation in interventions and outcomes, an in-depth scoping review is recommended to clarify research question viability would likely be more beneficial at this time. Given the variation in context, a realist review approach may also be suitable. |
| Mental Health (n = 34) 2000–2022 | US (19), Aus (3), India (2), UK (2), Chile (1), Ethiopia (1), Ireland (1), Japan (1), Lesotho (1), Mali (1), Rwanda (1), South Africa (1) | Service change (22) Patient education +/- other strategies (6) Provider training (6) | RCT (5) CS (4) US (25) | Outcomes varied by condition and aim. Depression symptoms and quality of life were common. | 34 studies assessed interventions that aimed to improve at least one patient clinical outcome associated with mental health. 44% of the studies focused on depression +/- anxiety, 24% focused on a broader range of multiple mental health conditions and 3 or fewer studies focused on ADHD, autism or suicide risk. There was a wide variety of interventions used. Service changes were the most common with collaborative care or adding a new service delivered by a nurse or mental health worker being among the most common. 15% used an RCT design and a further 9% used a control group the remainder were uncontrolled studies. | **In-depth scoping review.** This area shows promise for a *systematic review and potential meta-analysis*. However due to the wide variation in population, interventions and outcomes, an in-depth scoping review is recommended to clarify exact PICO questions. Given the variation in context, a realist review approach may also be suitable. |
| Hypertension (n = 13) 1997–2021 | US (7), China (2), Canada (1), Taiwan (1), Zambia (1), UK (1) | Patient education +/- other strategies (6) Provider training only (1) Service change (6) | RCT (4) CS (2) US (7) | Blood pressure (11) Blood pressure control (1) Stroke risk (1) | Thirteen studies assessed interventions that aimed to improve at least one patient clinical outcome associated with hypertension (e.g. blood pressure). Patient education was the most common type of intervention tested. 31% used an RCT design and a further 15% used a control group, the remainder were uncontrolled. | **In-depth scoping review** While all studies assessed a common outcome of blood pressure, the multicomponent heterogeneous nature of most interventions may suggest an in-depth scoping review is warranted to confirm feasible research questions for a systematic review or realist review. |

(*Continued*)

**Table 5.** (*Continued*)

| Topic Study (n) Year Range | Countries, Year Range | Intervention | Study Designs | Outcome | Content Summary | Suggestions for future research |
|---|---|---|---|---|---|---|
| CVD (n = 12) 2000–2019 | US (2), UK (2), China (2), Aus (1), India (1), Indonesia (1), New Zealand (1), Poland (1), Sweden (1) | Patient education +/- other strategies (6) Provider training only (1) Service change (5) | RCT (2) CS (4) US (6) | CHO (5) Blood pressure (8) CVD risk (4) | Twelve studies assessed interventions that aimed to improve at least one patient clinical outcome associated with CVD (e.g. CHO, BP or CVD risk). Half of the studies used patient education and the remainder used different types of changes in service structure. 17% of the studies used a RCT design and a further 33% used a control group, the remainder were uncontrolled studies. | **Systematic review (with realist review approach)** This area shows promise for a systematic review with a potential meta-analysis of patient education interventions and narrative synthesis of other service change interventions may be warranted. Additionally, a realist review approach may be used to explore impact of context. |
| Weight mgmt. (n = 9) 2008–2022 | US (8), Aus (1) | Patient education +/- other strategies (7) Service change (2) | RCT (6) US (2) | weight change (6) BMI (3) Unclear (2) | Nine studies assessed interventions that aimed to improve at least one patient clinical outcome associated with weight management (e.g. change in weight and BMI). The most common intervention type followed distantly some type of change in service structure. 67% of the studies used a RCT design and the remaining studies were uncontrolled. | **Systematic review.** This area shows promise for a systematic review and a potential meta-analysis for patient education interventions for weight loss. |
| Cancer (n = 8) 1998–2017 | US (3), Aus (3), Nepal (1), Sweden (1) | Service change (5) patient or provider education (2), audit and feedback (1) | RCT (4) US (4) | Varied | Eight studies assessed interventions that aimed to improve at least one patient clinical outcome associated with cancer. However, within this category, there were five different cancer types being studied (breast, cervical, colorectal, skin and multiple types) and the outcomes were also quite varied from cancer awareness to screening to patient satisfaction and quality of life. Interventions also varied with no single intervention type used in more than 2 studies. 50% of studies used a RCT design and the remainder were uncontrolled. | **In-depth scoping review.** While this area did have high quality study designs, the heterogeneity in cancer types, interventions and outcomes make it hard to recommend any specific question for a systematic review. This area may benefit from a more in-depth scoping review to confirm questions for a systematic review. |

(*Continued*)

**Table 5.** (Continued)

| Topic Study (n) Year Range | Countries, Year Range | Intervention | Study Designs | Outcome | Content Summary | Suggestions for future research |
|---|---|---|---|---|---|---|
| Pain Mgmt (n = 8) 2001–2020 | US (5), Canada (2), UK (1) | Patient education (5) Services change (3) | RCT (2) CS (3) US (3) | Pain intensity (2) Disability (4) Medication (2) | Eight studies assessed interventions that aimed to improve at least one patient clinical outcome associated with pain mgmt. (pain intensity or disability or changes in medication). Patient education was the most common intervention type and the remainder used different types of changes to service structure. 25% used a RCT design and a further 38% had a control group with the remainder of studies uncontrolled. | **Systematic Review** A formal systematic review and potential meta-analysis may be warranted for patient education interventions. |
| HIV (n = 7) 2006–2020 | South Africa (2), China (1), Ethiopia (1), Nigeria (1), Swaziland (1), US (1) | Service changes (7) | RCT (2) US (5) | CD4 counts (2) Viral load (1) or suppression (1) Unclear (3) | Seven studies assessed interventions that aimed to improve at least one patient clinical outcome associated with HIV, however, the outcomes were varied, with CD4 counts, viral load or suppression being the common clinical outcomes. Interventions also varied with most involving some change to service structure, however, no single intervention type was used in more than 2 studies. 29% of studies used a RCT design and the remainder were uncontrolled. | **Additional research** More high quality studies HIV outcomes for patients is likely warranted, however, this may not be feasible and thus an in-depth scoping review with a content expert may also be warranted to help guide future research questions and where higher quality studies may be needed or not. |
| Asthma (n = 6) 2001–2017 | US (3), Aus (2), Canada (1) | Provider training (3) Patient education (2) Service change (1) | CS (1) US (5) | Asthma control (3) FEV1% (3) | Six studies assessed interventions that aimed to improve at least one patient clinical outcome related to asthma (). 50% used provider education, 33% used patient education and 17% used some type of service structure change. No studies used a RCT design, 1 study used a control group with the remainder of studies uncontrolled. | **Additional research** Higher quality studies are likely needed but are also likely impractical. Thus, a formal scoping review or systematic review with narrative synthesis may be warranted instead. |
| Chronic Disease (n = 4) 2009–2018 | Canada (2), China (1), India (1) | Service change (3) Patient education (1) | RCT (2) CS (1) US (1) | Varied. QoL & Disease risk were common | 4 studies assessed interventions that aimed to improve an aspect of care related to chronic disease. Different combinations of multiple chronic conditions and interventions were seen across studies. 50% of studies used a RCT design, 1 study used a control group and 1 was uncontrolled. | **Additional research** |

(*Continued*)

**Table 5.** (Continued)

| Topic Study (n) Year Range | Countries, Year Range | Intervention | Study Designs | Outcome | Content Summary | Suggestions for future research |
|---|---|---|---|---|---|---|
| Vaccination (n = 4) 2001–2016 | US (2), China (1), UK (1) | Patient education +/- other strategies (2) Service change (2) | RCT (2) US (2) | Vaccination rate (4) | Four studies assessed interventions that aimed to improve at least one patient clinical outcome related to vaccinations (e.g. vaccinations rates). Studies were a mixture of patient education and service changes. 50% used a RCT design and two were uncontrolled. | **Systematic review** A systematic review and meta-analysis may be possible for the RCTs alongside a narrative synthesis of lower quality study designs. |
| Alcohol use (n = 3) 2007–2010 | Aus (1), US (1), Thailand (1) | Patient education (3) | RCT (3) | Drinks per day (3) | Three studies assessed interventions that aimed to improve at least one patient clinical outcome associated with alcohol use (drinks per day). All used patient education as the interventions and all used a RCT design. | **Systematic review** Systematic review with potential meta-analysis. A realist approach may be warranted given the different contexts in which the interventions were tested. |
| COPD (n = 3) 2009–2019 | China (1), Greece (1), US (1) | Provider training (2) Patient education (1) | RCT (1) US (2) | Unclear | Three studies assessed interventions that aimed to improve at least one patient clinical outcome associated with COPD. A range of outcomes were assessed related to diet, exercise, air flow obstruction and COPD general questionnaires but there was little similarity across studies. Only one used a RCT design and the remainder were uncontrolled. | **Additional research** Higher quality studies seem feasible for this type of health condition and thus, it is likely that the evidence base would benefit from investment in higher quality studies to improve outcomes for this condition. |
| Palliative Care (n = 3) 2016–2018 | Aus (2), US (1) | Service change (3) | US (3) | Unclear | Three studies assessed interventions that aimed to improve at least one patient clinical outcome associated with palliative care but no study assessed the same outcomes. Different types of service changes were used in each study and all studies were uncontrolled. | **Systematic review (narrative synthesis)** |
| Pregnancy related (n = 3) 2018–2019 | Ethiopia (1), Dominican Republic (1), Kenya (1) | Service change (3) | US (3) | Unclear | Three studies assessed interventions that aimed to improve at least one patient clinical outcome associated with pregnancy. All were conducted in low income countries and were uncontrolled studies. all interventions appear to use a change to service structure but in different ways. | **Additional research** Higher quality studies are likely needed but are also likely impractical. Thus, a formal scoping review or systematic review wit narrative synthesis may be warranted instead. |

(*Continued*)

**Table 5.** (Continued)

| Topic Study (n) Year Range | Countries, Year Range | Intervention | Study Designs | Outcome | Content Summary | Suggestions for future research |
|---|---|---|---|---|---|---|
| Sexually transmitted infections (n = 3) 2018–2019 | Aus (3) | Service change (3) | RCT (3) | Prevalence (2) Unclear (1) | Three studies assessed interventions that aimed to improve at least one patient clinical outcome associated with sexually transmitted infections (either prevalence or diagnostic test accuracy). All were conducted in Australia and all used a RCT design. | **Systematic review** The interventions may not be similar enough for meta-analysis. A narrative synthesis may be warranted. |
| Stroke Rehab (n = 3) 2004–2021 | China (1), India (1), Norway (1) | Service change (3) | RCT (3) | Function (2) Secondary Prevention (1) | Three studies assessed interventions that aimed to improve at least one patient clinical outcome associated with stroke rehab (function and prevention of second stroke). All used a type of service change most including some type of home rehab plus other types of strategies as the intervention and all used a RCT design. | **Systematic review** The interventions may not be similar enough for meta-analysis. A narrative synthesis may be warranted. A realist approach may be warranted given the different contexts in which the interventions were tested. |

**Notes**:

*Study design*: RCTs include individual and cluster RCTs, CS include non-randomized studies with a control group (i.e. controlled before-after studies and interrupted time series, US includes all studies without a pre-assigned control group (uncontrolled before-after, cohort studies or cross sectional studies)

*Interventions*: Due to the many types of interventions that could not all be easily listed in this table, "service change" was used to capture any of the intervention codes which involved changing some aspect of the health system (extended scope of practice, telehealth/virtual care, co-ordination/referral pathways, reorganization of service or implementation of a new service)

*Data not included in this table*: *Health topics that were assessed in 2 or less studies include*: Atrial fibrillation, burn rehab, Chagas disease, childhood diarrhea, dementia, epilepsy, kidney disease, lung Disease, osteoarthritis, osteoporosis, smoking cessation, speech impairments, vision impairments. *Topic areas without a specific health topic*: Two health areas did not include a specific topic and were not included for this content summary: general health and medication related topics

diarrhea, dementia, epilepsy, kidney disease, lung disease, osteoarthritis, osteoporosis, smoking cessation, speech impairments, vision impairments.

**Patient experience (n = 34).** Most of the 34 articles aiming to improve quality of care by enhancing patient experience were published between 2010–2022. The majority of the studies were published in the United States (n = 19). The most commonly used interventions involved reorganizing existing services to optimize care (n = 6) or expanding the scope of practice of non-family physician healthcare providers (n = 5). Some of the interventions were used in combination as well (e.g., provider education/training + audit and feedback). The most common health topics included mental health (n = 6) and pregnancy (n = 3). Additionally, patient experience related to care for minor illness, burn rehab, cancer, chronic disease, diabetes, hypertension, and pain was studied by 2 or less studies. Most studies used uncontrolled before-after designs (n = 9), randomized controlled trials (n = 8), or cross-sectional designs (n = 7). Patient experience was typically measured via questionnaires, interviews, and surveys. Table 6 provides a content summary analysis for the two health topics that were assessed in three or more studies. Among these, we judged mental health (n = 6) to be ready for systematic review. Additional research is required for pregnancy (n = 3) as all three identified studies

**Table 6. Content summary for quality—Patient experience (health topics that were assessed by 3 or more studies are reported in the content summary).**

| Topic Study (n) | Countries, Year Range | Intervention | Study Design | Experience outcome measured? Yes/No/Unclear* Type | Content Summary | Suggestions for future research |
|---|---|---|---|---|---|---|
| Mental Health (n = 6) 2007–2018 | US (4) Chile (1) UK (1) | Service Changes (3) Patient education (2) Provider training (1) | RCT (1) US (5) | Yes (6) satisfaction | Seven studies assessed outcomes related to patient experience (i.e. satisfaction with care/service) mostly in the US. The interventions varied but a majority used some form of a collaborative care model and the remainder implemented a form of psychological education such as CBT. Only one study used a randomized design and the remainder were uncontrolled. | **Systematic review** There appears to be sufficient data assessing patient satisfaction with service changes that involve some form of collaborative care that would warrant a systematic review. |
| Pregnancy (n = 3) 2010–2018 | Australia (1) Canada (1) Ethiopia (1) | Service Changes (3) | US (3) | Yes (3) Satisfaction (1) Needs met (1) Overall experience (1) | Three studies assessed outcomes related to patient experience in three different countries. The interventions were all different but were about implementing a new model of care for pregnant women. All studies assessed different types of experience measures all were uncontrolled studies. | **More research** Each of the three studies assessed a different aspect of patient experience with three different types of new care models in three different countries. This could be included as part of a larger scoping review on the health topic of pregnancy related issues but would be sufficient to warrant its own review topic. |

**Notes**:

*Unclear outcomes**: this means that the study suggested it measured the outcome but it wasn't clear from the methods or results how this was actually assessed and follow-up with study authors would be required to confirm if this outcome was assessed.

*Study design*: RCTs include individual and cluster RCTs, CS include non-randomized studies with a control group (i.e. controlled before-after studies and interrupted time series, US includes all studies without a pre-assigned control group (uncontrolled before-after, cohort studies or cross sectional studies)

*Interventions*: Due to the many types of interventions that could not all be easily listed in this table, "service change" was used to capture any of the intervention codes which involved changing some aspect of the health system (extended scope of practice, telehealth/virtual care, co-ordination/referral pathways, reorganization of service or implementation of a new service)

*Data not included in this table*:

*Health topics that were assessed in 2 or less studies include*:

Antibiotic prescribing, burn rehabilitation, cancer, chronic disease, diabetes, hypertension, pain management, palliative care, stroke rehabilitation, weight management.

*Topic areas without a specific health topic*:

Any condition, medication related, minor illness.

were assessing different interventions and different outcomes. Additional research is recommended for this outcome category as a whole since there were so few studies examining this outcome.

**Equity of care (n = 4).** Four studies examining equity of care were published between 2010–2022 including one study each from Ethiopia, India, Kenya, and the United States. The interventions from these studies involved reorganization of services (n = 2), provider training (n = 1), and a combination of provider training and lay community member training (n = 1). Three studies focused on mental health and one on pregnancy. Table 7 provides a content summary analysis for the health topic that was assessed in three or more studies. Of note, several papers did not assess a specific health topic (e.g., they were studying an intervention applicable to any condition). We judged the topic of mental health (n = 3) to require additional research; two studies assessed similar outcomes but there was not enough similarity among interventions to warrant a review. Additional research is recommended for this outcome category as a whole since there were so few studies examining this outcome.

**Patient safety (n = 1).** Only one study conducted in 2019 from the United States examined patient safety. The intervention involved coordinating care to improve medication

**Table 7. Content summary for quality—Equity (health topics that were assessed by 3 or more studies are reported in the content summary).**

| Topic Study (n) | Countries, Year Range | Intervention | Study Design | Experience outcome measured? Yes/No/ Unclear* Type | Content Summary | Suggestions for future research |
|---|---|---|---|---|---|---|
| Mental Health (n = 3) 2019–2020 | US (1) Ethiopia (1) Kenya (1) | Service Changes (3) Patient education (2) Provider training (1) | US (3) | Yes (3) Discrimination (2) Universal screening (1) | Three studies assessed outcomes related to equity of care. All studies were conducted in different countries. Two studies assessed stigma and experience of discrimination and one study assessed provision of depression screening for all appropriate patients. Two studies used interdisciplinary team-based care to deliver mental health services and one study used provider training on mental health knowledge and awareness. All studies were uncontrolled. | **More research** While two studies assessed discrimination there was not enough similarity amongst interventions to warrant a systematic review. |

**Notes**:

*Unclear outcomes**: this means that the study suggested it measured the outcome but it wasn't clear from the methods or results how this was actually assessed and follow-up with study authors would be required to confirm if this outcome was assessed.

*Study design*: RCTs include individual and cluster RCTs, CS include non-randomized studies with a control group (i.e. controlled before-after studies and interrupted time series, US includes all studies without a pre-assigned control group (uncontrolled before-after, cohort studies or cross sectional studies)

*Interventions*: Due to the many types of interventions that could not all be easily listed in this table, "service change" was used to capture any of the intervention codes which involved changing some aspect of the health system (extended scope of practice, telehealth/virtual care, co-ordination/referral pathways, reorganization of service or implementation of a new service)

*Data not included in this table*:

*Health topics that were assessed in 2 or less studies include*:

Pregnancy-related.

reconciliation in a primary care clinic. Additional research is recommended for this outcome category as a whole since there were so few studies examining this outcome.

## Access to care

Seventy-six (20%) articles focused on interventions to improve access to primary care. Among these articles, 21 (28%) focused on recruitment of family physicians, 15 (20%) on retention of family physicians, 14 (18%) on recruitment of alternative health provider to provide family physician-related services, and 35 (46%) on access to other services not typically provided in primary care. Several of these studies assessed more than one outcome. A detailed breakdown of all intervention types, study designs, individual study outcomes, and outcome measurements, for access to care studies can be found in S8–S11 Appendices. This data is summarized in the following sections.

**Access to a primary care provider (n = 42).** Forty-two articles focused on improving access to primary care providers. The articles were largely published between 2010–2021 and came from 13 different countries, with the majority originating from the United States (n = 19) and Australia (n = 7). The most common study designs were cohort studies (n = 19), uncontrolled before-after studies (n = 10) and controlled before-after studies (n = 7). We found three main provider access topics, all of which were assessed in 3 or more studies, including recruitment (n = 21) and/or retention (n = 15) of family physicians, and recruitment of alternative primary care providers (e.g., nurse practitioners, pharmacists) (n = 14). Across the topics, different types of interventions were used, the most common type of intervention involved extending the scope of practice of non-GPs (n = 13), and the most common types of intervention for recruiting and retaining family physicians were implemented during medical

education via selective recruitment of students with a focus on rural practice into medical education (n = 10), or medical education interventions that involved exposure to rural practice (n = 9), followed by financial incentives to assist with tuition payback (n = 6). A content summary for each of the three topic areas can be found in Table 8. We judged all three topics to be ready for systematic review.

**Access to services (n = 36).** The majority (77%) of the 36 studies that aimed to increase access to primary care services conducted between 2010–2022. These were conducted in the United States (n = 12), Australia (n = 3), Canada (n = 2), India (n = 2), and multiple other countries. The most commonly used interventions involved providing training to existing health professionals to deliver different types of additional services they would not normally deliver (n = 13) or training lay community members (n = 4). Other studies also commonly aimed to increase referral to existing services by improving awareness of services or ease of referral (n = 6). Most common study designs were uncontrolled before-after (n = 13), cohort designs (n = 8), cross-sectional (n = 6), interrupted time series (n = 4), and randomized controlled trials (n = 3). We identified three service area topics that were assessed in three or more studies including diabetes (n = 5), mental health (n = 4) and pregnancy (n = 4). A content summary for each health topics can be found in Table 9. Of note, several papers did not assess a specific health topic (e.g., they were studying an intervention applicable to any condition). For each of these topic areas, there was a wide range of interventions and outcome measurement methods. We judged all topics to be amenable to an in-depth scoping review.

## Efficiency of care delivery

Only 47 studies (15% of the total) examined efficiency of care. Among these, 28 (60%) included interventions aimed at improving cost-savings, 26 (55%) included interventions to improve healthcare use, and three included an intervention aimed at improving patient wait times. A detailed breakdown of all intervention types, study designs, individual study outcomes, and outcome measurements for efficiency studies can be found in Table 4. This data is summarized in the following paragraphs.

**Wait times (n = 3).** Three studies conducted in 2008, 2017, and 2020 in the United States (n = 2) and Canada (n = 1) assessed wait times. The interventions involved improving coordination/referral pathways, reorganization of services, and provider training. Two studies used an uncontrolled before-after study design and one used a prospective cohort design. Two studies focused on mental health and one on surgery. Wait times for new appointments were measured using data from a computerized monitoring system and electronic medical record data. Additional research is recommended for this outcome category as a whole as there were so few studies examining this outcome.

**Healthcare utilization (n = 26).** Over 80% of the 26 articles assessing healthcare utilization were published between 2010–2022. Most of these studies were conducted in the United States (n = 12), Canada (n = 4), Australia (n = 3), and South Africa (n = 2). The most common intervention included reorganization of services to optimize care delivery (n = 7) and improving coordination/referral pathways (n = 5). Some studies used a combination of interventions (e.g., expanding scope of practice + provider training/upskilling). The majority of the studies used a controlled before-after (n = 8), an uncontrolled before-after (n = 7), or cohort (n = 5) design. The most common health topics assessed were mental health (n = 2) and palliative care (n = 2). Additional research is recommended for this outcome category as a whole since it was not assessed in relation to the same health topic for three or more studies.

**Costing and cost-effectiveness (n = 28).** Most of the 28 studies examining cost-savings as a means to improve efficiency were published between 2010–2022. These were conducted in a

**Table 8. Content summary for access—To a primary care provider (access topics that were assessed by 3 or more studies are reported in the content summary).**

| Access topic Study (n) | Countries, Year Range | Intervention | Study Designs | Outcome | Content Summary | Suggestions for future research |
|---|---|---|---|---|---|---|
| Recruitment of FPs (n = 21) | US (13), Aus (2), France (2), Norway (2), CAN (1), Germany (1) 2001–2021 | Medical education—exposure (8) Medical Education–selective recruitment (8) Financial incentives (4) Service change (1) | CS (3) US (18) | # graduates practicing in a rural/ remote community upon graduation | Twenty-one studies tested interventions to increase recruit family physicians in rural or remote communities. There were 4 different types of interventions, the most common were changes made during the medical school training process either by exposure to rural areas during training or only recruiting applicants who lived or grew up in a rural or remote location and had a firm commitment to practicing in that area after training. Another 4 studies used some form of financial incentive mostly via medical school fee payment. Lastly, one study used a service change by implementing multidisciplinary family care teams to support collaboration and better working conditions to try and entice recruitment to rural or remote locations. Most studies were uncontrolled. | **Systematic review (realist approach)** A systematic review of these three different strategies to increase access to primary care providers in rural areas. There have been reviews of this topic, however most have focused on family physicians and none provided a comprehensive review of the effectiveness of these different strategies with an understanding of implications for decision makers. Additionally, while RCTs or controlled trials may be hard to design for this topic area, a formal systematic review can provide thorough assessments of study quality and specific directions for future research to advance this area. |
| Retention of FPs (n = 15) | US (6), Aus (3), Norway (1), Japan (2), Can (1), France (1) 2001–2020 | Medical education—exposure (5) Medical Education–selective recruitment (4) Financial incentives (4) Well-being (2) | CS (4) US (11) | working in rural area up to 16 years post-grad (11) GP density (1) Intention to continue rural work (3) | Fifteen studies tested interventions to increase retention of family physicians in rural or remote communities. There were 4 different types of interventions, many similar to those issued for recruitment: the most common was exposure to rural areas during medical training which varied in duration and setting. Selective recruitment into medication school based on if the applicant lived or grew up in a rural or remote location and had a firm commitment to working in that area was used in 4 studies. Another 4 studies used some form of financial incentive mostly via medical school fee payment. Lastly, two studies used a well-being support program to try and increase retention in rural or remote locations. Both interventions offered social and emotional support but were based on different principals and delivered in different ways. Most studies were uncontrolled. | |
| Adding different types of PC providers (n = 14) 2000–2019 | US (4), UK (3), Aus (2), Croatia (1), Dominican Republic (1), India (1), South Africa (1), Taiwan (1) | Extended scope of practice (13) Training of lay community members (1) | RCT (2) CS (1) US (11) | Coverage or reach via patient use (5) Exact measure unclear (9) | 14 studies tested interventions to increase access to an alternative primary care provider. The interventions included extending the scope of practice of health professionals including Nurses (6), Pharmacists (3), mental health workers (1), residents (1), and community or lay village health workers (2). The extended scope of practice related to a variety of heath topics (e.g. mental health (3), medications (2), minor illness (2), diabetes (1)). Two studies used an RCT design and a further study used a control group and the remainder were uncontrolled. | |

**Notes**:

*Study design*: RCTs include individual and cluster RCTs, CS include non-randomized studies with a control group (i.e. controlled before-after studies and interrupted time series, US includes all studies without a pre-assigned control group (uncontrolled before-after, cohort studies or cross-sectional studies)

*Interventions*: Due to the many types of interventions that could not all be easily listed in this table, "service change" was used to capture any of the intervention codes which involved changing some aspect of the health system (extended scope of practice, telehealth/virtual care, co-ordination/referral pathways, reorganization of service or implementation of a new service)

**Table 9. Content summary access—To a primary care services (access topics that were assessed by 3 or more studies are reported in the content summary).**

| Access topic Study (n) | Countries, Year Range | Intervention | Study Designs | Outcomes related to coverage via referral volume, patient use/ attendance | Content Summary | Suggestions for future research |
|---|---|---|---|---|---|---|
| Diabetes (n = 5) | US (4), Spain (1) 2009–2020 | Different types of Service change (5) | CS (2) US (3) | Yes (2) Unclear (3) | Five studies tested interventions to increase access to primary care by offering extra services related to Diabetes management and screening for complications. Interventions were quite varied in nature, some related to extra education clinics or home visits, some were implementing a new service for retinopathy or neuropathy screening and one used clinic-based quality improvement to offer additional services where needed. Two studies used a control group but 3 were uncontrolled. | **Scoping review** For each of these topic areas, there was a wide range of interventions and outcome measurement methods. An in-depth scoping review that focuses on how extra services are used to increase access to primary care in rural areas is warranted. |
| Mental Health (n = 4) | US (3), India (1) 2008–2022 | Different types of Service change (4) | CS (1) US (3) | Yes (3) Unclear (1) | Four studies tested interventions to increase access to primary care by offering extra services for mental health (e.g.). Interventions included coordination efforts such as collaborative care and task sharing or education clinics. Most studies were uncontrolled. | |
| Pregnancy related (n = 4) | Burkina Faso (1), Dominican Republic (1), Ghana (1), Kenya (1) 2016–2019 | Different types of Service change (4) | US (4) | Yes (2) Unclear (2) | Four studies tested interventions to increase access to primary care by offering extra services related to pregnancy. The interventions included implementing a new service and in some cases extending the scope of practice of existing health professionals or lay community members to offer the extra services. all studies were uncontrolled | |

Notes:

*Study design*: RCTs include individual and cluster RCTs, CS include non-randomized studies with a control group (i.e. controlled before-after studies and interrupted time series, US includes all studies without a pre-assigned control group (uncontrolled before-after, cohort studies or cross-sectional studies)

*Interventions*: Due to the many types of interventions that could not all be easily listed in this table, "service change" was used to capture any of the intervention codes which involved changing some aspect of the health system (extended scope of practice, telehealth/virtual care, co-ordination/referral pathways, reorganization of service or implementation of a new service)

*Data not included in this table*:

*Health topics that were assessed in 2 or less studies include*: cancer, chronic disease, CVD, febrile illness, HIV, Malaria, maternal and child health, NCD, speech impairments.

*Topic areas without a specific health topic*: three areas did not include a specific health topic: general health, medication related, and preventative care.

variety of countries, most commonly in the United States (n = 12). The majority of the interventions focused on reorganizing services to optimize care delivery (n = 8), improving coordination/referral pathways (n = 6), and upskilling healthcare providers on evidence-based clinical practice guidelines (n = 4). The most common study design in this area was the uncontrolled before-after design (n = 10). Mental health (n = 3), preventive care (n = 2), CVD (n = 2), and epilepsy (n = 2) were the most commonly assessed health topics. A content summary for mental health can be found in Table 10. Of note, several papers did not assess a

**Table 10. Content summary efficiency—Costing and cost-effectiveness (efficiency topics that were assessed by 3 or more studies are reported in the content summary).**

| Topic Study (n) | Countries, Year Range | Intervention | Study Design | Efficiency outcome measured? Yes/No/ Unclear* | Content Summary | Suggestions for future research |
|---|---|---|---|---|---|---|
| | | | | Cost | | |
| Mental Health (n = 3) 2015–2017 | US (3) | Provider training (2) Service Changes (1) | RCT (1) US (2) | Yes (3) | Three studies assessed cost-related outcomes of implementing a new or change in service for improving mental health outcomes (e.g. adherence to guidelines, depression, quality of life). Costing outcomes varied across studies with only one study measuring cost-effectiveness with QALYs. | In-depth scoping review. While three studies assessed cost outcomes related to the topic area of mental health, the interventions and outcomes were different across studies precluding a specific PICO question. |

**Notes**:

*Unclear outcomes**: this means that the study suggested it measured the outcome, but it wasn't clear from the methods or results how this was actually assessed and follow-up with study authors would be required to confirm if this outcome was assessed.

*Study design*: RCTs include individual and cluster RCTs, CS include non-randomized studies with a control group (i.e. controlled before-after studies and interrupted time series, US includes all studies without a pre-assigned control group (uncontrolled before-after, cohort studies or cross-sectional studies)

*Interventions*: Due to the many types of interventions that could not all be easily listed in this table, "service change" was used to capture any of the intervention codes which involved changing some aspect of the health system (extended scope of practice, telehealth/virtual care, co-ordination/referral pathways, reorganization of service or implementation of a new service)

*Data not included in this table*:

*Health topics that were assessed in 2 or less studies include*:

Atrial fibrillation, cancer, chronic, CVD, diabetes, epilepsy, febrile illness, osteoporosis, palliative care, respiratory tract infection, osteoarthritis, dental care.

*Topic areas without a specific health topic*:

specific health topic (e.g., they were studying an intervention applicable to any condition). We recommend an in-depth scoping review for one health topic (mental health, n = 3). Additional research is also recommended for this outcome category as a whole as there were relatively few studies examining this outcome overall.

## Discussion

This review serves as an overarching scoping review that primarily aims to identify what types of topics, interventions and designs are being used to improve access, quality and efficiency of primary care in rural and remote areas globally. A secondary aim is to provide a high-level content summary within each of these outcome areas and to provide recommendations for future research. We have separated our discussion of the results into two main sections, (i) content summary and recommendations for future research which includes a summary of the main findings to answer our objectives, and (ii) the burden of rurality and the need for rural-specific research which provides a thoughtful discussion of these results from the rural clinical perspective.

### Content summary and recommendations for further research

Overall, given our liberal search criteria, we found a low number of papers that met our inclusion criteria and an even lower number that used more rigorous randomized designs. While we did note an upward trend toward increased publications, the trend we observed peaked in 2017 before beginning a decline to pre-peak levels. In addition, the increase in the yearly number of rural primary care publications must be tempered by the fact that scientific output in

general is doubling approximately every nine years [391]. Nevertheless, the increase we observed in rural primary care literature does appear to outpace the rate of increase in general scientific productivity. Not surprisingly, a large percentage of the papers captured in this review originate from the United States, Australia and Canada, countries that have long been known to have high levels of rurality and established programs of rural health research. Notably, a substantial proportion of papers come from Africa (n = 35 (9.4%), Tables 2–4) despite limited resources in those countries. European, Middle Eastern and Asian countries are under-represented in this review, likely because of relatively high population density and a lower impact of rurality at the population level. In terms of our outcome areas of interest, we will present our findings summary for each separately.

Access: We found the topic area of increasing access to a primary care provider one that is ready for a formal systematic review. While each of the identified three subtopics, recruitment or retention of family physicians and recruitment of an alternate care provider, have been the subject of previous reviews there hasn't been a comprehensive review of the effectiveness of all available strategies which would be of benefit as this issue is a high priority in all areas. A realist approach may work best as there are several different health system contexts to consider in likelihood of effectiveness. The topic of increasing access to primary care services included a number of different services for specific health issues such as diabetes or mental health which would benefit from a further scoping review to identify useful PICO questions.

Efficiency: We found only 2 topics that assessed outcomes related to improving efficiency of primary care services, mental health and pregnancy-related topics. Unfortunately, reporting regarding the outcomes of wait-times or healthcare use were unclear and follow-up with authors would be necessary to understand these outcomes in more detail before meaningful PICOs could be identified.

Quality: We found the majority of studies aimed to improve quality of primary care either via interventions to improve adherence to guidelines (improving evidence-based medicine) or to improve specific patient clinical outcomes. Much less attention has been paid to other aspects of quality such as patient experience or equity. The most common topics across all areas of quality were the health topics of diabetes, mental health, cancer, CVD and asthma. In our results, we made several recommendations for where the evidence is ready for systematic or scoping reviews within the specific health topics, however, looking at the data across outcomes, we would also recommend broader reviews for the same or interrelated health topics. For example, the health topics of diabetes, cancer, asthma, CVD, mental health and pregnancy all had studies that assessed interventions to improve patient clinical outcomes and adherence to/update of evidence-based care recommendations. In these cases, we would suggest conducting comprehensive reviews by these topic areas that include all outcomes of interest that have been assessed. Additionally, we found evidence to suggest systematic reviews for a number of related health topics such as weight management, alcohol use and pain management. We would suggest combining these topics to form a larger topic area such as lifestyle changes for a more impactful review. The last comment worth note, is that a realist approach would likely be required in almost all reviews to assess the findings in terms of contextual factors related to (i) geographical location, (ii) country economic status and (iii) health system status (private/public, etc.)

## The burden of rurality and the need for rural-specific research

As we all know, there is a huge and rapidly growing literature on the management of chronic diseases and their burden on society and the health system. In most jurisdictions, rurality is associated with a 2–5 year deficit in life expectancy [2, 5, 392, 393] which does not compare to

the effect of widely studied diseases such as diabetes, heart disease and stroke [394]. However, the far greater prevalence of rurality suggests that its burden, at the population level, is likely to be as great or greater than any of the chronic diseases alone. Admittedly, some of the urban/rural disparity in health outcomes is attributable to differences in rates of chronic disease [3, 4]. Thus, science that examines chronic disease may indirectly contribute to narrowing the rural-urban divide. However, health services delivery and socio-cultural issues are different in rural areas, suggesting that rural-specific research may be necessary to more completely address rural/urban health inequities than disease inquiry alone.

For this work, we deliberately chose liberal inclusion criteria to maximize reach, by including designs such as non-randomized and before-after studies, which are often excluded from reviews. During article screening, we did not tally papers that were excluded because of the lack of a comparative evaluation component (e.g., opinion pieces, rural program descriptions and uncontrolled cross-sectional evaluations), but subjectively note that this was a large number of papers. This suggests that there is considerable interest in rural healthcare delivery not reflected in this review, but a lack of resources to pursue more intensive research efforts, mirroring the lack of resources available to health service delivery in rural areas.

As we have noted in Table 2, the bulk of papers included in this review examined the quality of care delivery. We have not attempted to systematically identify whether the interventions studied in these papers were specifically tailored for rural populations or simply studied a geography-agnostic program delivered in a rural area. However, our subjective opinion is that a substantial proportion of the studied interventions did not include modifications for rural populations and provided only limited if any description of factors that may have influenced program implementation, delivery or quality specifically in the rural setting. Nearly a third of the papers examined the adoption of best practices, most of which were drawn from clinical practice guidelines which typically do not include a rural focus. Programs that did include both urban and rural programs often included only a small number of sites, limiting the generalizability of the findings. Given relatively limited attempts to create rural-specific data analyses or amendments to interventions in the quality of care papers, the contribution to the rural-specific primary care literature is somewhat less than suggested by the high number of papers in this section. This represents an important gap in the literature that could be filled by a greater number of studies that examine differences in implementation, delivery and outcomes between rural and urban sites of the same program and/or a systematic review of studies that include both rural and urban sites to elucidate factors (if any) that contribute to differences in outcomes between rural and urban populations. The results of this work could then be used to develop frameworks describing which evidence from primarily urban sites could be applied in the rural setting, and where clinical programs require modification for rural delivery.

Reflecting the challenges with and widespread attention paid to the recruitment and retention of primary care physicians as a means of overcoming one of the greatest challenges in rural care delivery [395, 396], our paper documents a considerable number of papers (n = 28) that examined these issues, but none that used randomized designs. However, given the potential for multi-disciplinary providers to meet primary care needs, it is surprising that relatively few papers (n = 14) examined the use of non-physician providers in the rural setting. In the broad primary care literature (i.e. not exclusively rural), there is widespread interest in multi-disciplinary primary care delivery, and many of the findings from that literature could be used to inform rural service delivery. However, as mentioned above, socio-cultural and other issues are different in rural areas, which may lead to service-delivery challenges not seen with urban teams. This issue further highlights the need for studies to determine the extent to which evidence developed in urban centres can be applied in the rural setting.

## Limitations

Given the high number of papers included in this scoping review, we have not attempted to extract specific recommendations regarding policy or service delivery, but we intend to do so in follow-up sub-topic papers. As with any review, the quality is limited by the quality of the literature search and data extraction, but we have used an experienced information specialist and duplicate data extraction approaches to mitigate these concerns. We have used the Canadian Foundation for Healthcare Improvements' 2012 policy document, entitled Toward a Primary Care Strategy for Canada to inform many of the working definitions for our review but we acknowledge that other frameworks and definitions exist to describe primary care and related outcomes and thus there may be some overlap or slight differences in our categorization compared to the literature [9]. Finally, in an attempt to describe the rural primary care literature as thoroughly as possible, we have classified papers into multiple categories if appropriate. Thus, interpretations about the degree of focus on a particular subject area should keep this in mind.

## Supporting information

**S1 Checklist. Preferred Reporting Items for Systematic reviews and Meta-Analyses extension for Scoping Reviews (PRISMA-ScR) checklist.**
(DOCX)

**S1 Appendix. Sample search strategy.**
(PDF)

**S2 Appendix. Details on article characteristics.**
(DOCX)

**S3 Appendix. Evidence-based practice.**
(DOCX)

**S4 Appendix. Clinical outcomes.**
(DOCX)

**S5 Appendix. Patient experience.**
(DOCX)

**S6 Appendix. Equity of care.**
(DOCX)

**S7 Appendix. Patient safety.**
(DOCX)

**S8 Appendix. Recruitment.**
(DOCX)

**S9 Appendix. Retention.**
(DOCX)

**S10 Appendix. Access to alternative primary care provider.**
(DOCX)

**S11 Appendix. Access to services not normally delivered by family physicians.**
(DOCX)

**S12 Appendix. Wait times.**
(DOCX)

**S13 Appendix. Healthcare utilization.**
(DOCX)

**S14 Appendix. Costing and cost effectiveness.**
(DOCX)

## Acknowledgments

The authors wish to thank Ms. Lindsay Alcock for assistance with refining the search strategy and completing the literature search.

## Author Contributions

**Conceptualization:** Kris Aubrey-Basler, Andrea Pike, James Rourke, Shabnam Asghari, Amanda Hall.

**Data curation:** Krystal Bursey, Carla Penney, Bradley Furlong, Mark Howells, Harith Al-Obaid, Amanda Hall.

**Formal analysis:** Krystal Bursey, Andrea Pike, Carla Penney, Mark Howells, Harith Al-Obaid, Amanda Hall.

**Funding acquisition:** Kris Aubrey-Basler.

**Methodology:** Kris Aubrey-Basler, Krystal Bursey, Amanda Hall.

**Project administration:** Andrea Pike, Amanda Hall.

**Supervision:** Kris Aubrey-Basler, Amanda Hall.

**Writing – original draft:** Kris Aubrey-Basler, Krystal Bursey, Andrea Pike, James Rourke, Shabnam Asghari, Amanda Hall.

**Writing – review & editing:** Kris Aubrey-Basler, Krystal Bursey, Andrea Pike, Carla Penney, Bradley Furlong, Mark Howells, Harith Al-Obaid, James Rourke, Shabnam Asghari, Amanda Hall.

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
