## [Decision Letter · Decision Letter 0]

8 Feb 2024

PONE-D-23-28342Interventions to improve primary healthcare in rural settings: A scoping reviewPLOS ONE

Dear Dr. Aubrey-Bassler,

Thank you for submitting your manuscript to PLOS ONE. After careful consideration, we feel that it has merit but does not fully meet PLOS ONE’s publication criteria as it currently stands. Therefore, we invite you to submit a revised version of the manuscript that addresses the points raised during the review process.

We look forward to receiving your revised manuscript.

Kind regards,

Jennifer Yourkavitch

Academic Editor

PLOS ONE

Journal Requirements:

   "KAB and JR are family physicians who previously practiced in rural areas. KAB, JR, and SA are rural health services researchers with an interest in the equity of health service distribution. The authors have no other disclosures to report."

4. We note that Figure 3 in your submission contain map/satellite images which may be copyrighted. All PLOS content is published under the Creative Commons Attribution License (CC BY 4.0), which means that the manuscript, images, and Supporting Information files will be freely available online, and any third party is permitted to access, download, copy, distribute, and use these materials in any way, even commercially, with proper attribution. For these reasons, we cannot publish previously copyrighted maps or satellite images created using proprietary data, such as Google software (Google Maps, Street View, and Earth). For more information, see our copyright guidelines: http://journals.plos.org/plosone/s/licenses-and-copyright.

a. You may seek permission from the original copyright holder of Figure 3 to publish the content specifically under the CC BY 4.0 license.  

5. We note that this manuscript is a systematic review or meta-analysis; our author guidelines therefore require that you use PRISMA guidance to help improve reporting quality of this type of study. Please upload copies of the completed PRISMA checklist as Supporting Information with a file name “PRISMA checklist”.

Additional Editor Comments:

Thank you for submitting this interesting manuscript. In general, we believe this treatment is too superficial. We understand that more in-depth treatment will be given to topics in subsequent papers but this paper should not merely be an appetizer. Readers should learn something from your effort. Try to give at least some high-level content analysis in the body of the article and more depth about the gaps and areas for further research. Please also respond to each comment from the reviewers.

Reviewers' comments:

Reviewer's Responses to Questions

**Comments to the Author**

1. Is the manuscript technically sound, and do the data support the conclusions?

Reviewer #1: Yes

Reviewer #2: Yes

2. Has the statistical analysis been performed appropriately and rigorously? 

Reviewer #1: N/A

Reviewer #2: N/A

3. Have the authors made all data underlying the findings in their manuscript fully available?

Reviewer #1: Yes

Reviewer #2: Yes

4. Is the manuscript presented in an intelligible fashion and written in standard English?

Reviewer #1: Yes

Reviewer #2: Yes

5. Review Comments to the Author

Reviewer #1: General comments

This is a scoping review of the literature on interventions to improve primary health care in rural settings. The review covers 1996-2022 and is wide-ranging. The authors do not limit their review to either high income countries (HICs) or low- and middle-income countries (LMICs), but include both. The review covers a variety of health conditions, outcomes (access; quality; efficiency) and study types. They excluded purely descriptive articles.

The authors made the decision to include articles from both HICs and LMICs. This, of course, means that the contexts vary widely in terms of the epidemiology of disease and the strength and organization of the health system. This makes for a more ambitious and difficult undertaking in terms of drawing conclusions.

Specific comments

Introduction

• The authors don’t clearly and explicitly state the definition or framework they are using for Primary Care or Primary Health Care. They seem to be screening articles mainly by level of care and excluding specialist or hospital-delivered care. But are they using any kind of characteristics of primary care, like the classic 4 C’s (first contact, coordination, comprehensiveness, and continuity). If the common feature of included articles is only that they are at the most basic level of care (i.e., “first contact”), but not necessarily that the care is coordinated or comprehensive, for instance, then are all included articles truly about “PHC” or “PC” or would they more accurately be described as about “basic health care” or “non-specialist care.” I take as an illustrative example of what I mean by this distinction one of the interventions listed in Appendix 3 on Clinical Outcomes. From what is listed in the table, it is not clear if this was a “vertically delivered” intervention that was not integrated into other health care. Perhaps it was integrated into a coordinated, comprehensive, and continuous system of care. But I don’t feel that there is enough information given for the reader to confirm that is the case.

* Shakya, 2016, Nepal

* Uncontrolled before/after

* The primary purpose of this study was to assess the knowledge of cervical cancer among women in rural Nepal and explore the feasibility and impact of a community-based awareness program on cervical cancer. Code: Patient Education/Navigation

* Community-based educational meetings on cervical cancer and its prevention were conducted among women’s groups in rural Nepal.

* Questionnaires with open-ended and closed-ended questions were administered through a face-to-face interview. All interviews were performed by the primary investigator and a doctor trained in interviewing. The questionnaire consisted of sociodemographic information (age, education level, and income source), personal information (age at marriage, number of live births, number of marriages), and questions on knowledge and attitudes related to cervical cancer to determine the outcome measures of interest. The messages that were emphasized during the education session were: (1) cervical cancer is preventable, (2) risk factors, signs, symptoms, and asymptomatic nature of early cervical cancer, and (3) the importance of undergoing gynaecological examination and cervical screening. The participants were asked whether they would participate in an upcoming cost-free cervical cancer screening program two weeks after the educational meeting.

Methods

• The authors followed a systematic and standard process for this Scoping Review, following the standard process originally outlined by Arksey and O'Malley in 2005 and the PRISMA-ScR checklist for presentation.

• The authors searched the following databases: PubMed, CINAHL, Cochrane Library, Embase, but did not attempt to include grey literature in their review. Can they give a reason for this?

• The authors present their search terms for an illustrative database (MEDLINE). I’m curious: It does not seem that the authors included any term on cost or cost-effectiveness although this is one of the sub-outcomes under Efficiency outcomes that they list in Table 3.

• In the PRISMA diagram the authors ought to list the reasons for exclusion (and how frequently the reason was used to exclude an article).

Results

• I am bit confused by the categorization of some issues as “outcomes” and others as strategies.” For instance, on Page 6, “recruitment of family physicians” and “retention of family physicians” are listed as sub-outcomes under Access, rather than as strategies or interventions that are meant to improve outcomes like availability.

• Related to the last point, I’m not sure how or why the authors decided to sub-divide the outcome of efficiency. A fairly well-known sub-division of this concept is the following: availability, accessibility, accommodation, affordability, acceptability (Penchansky R, Thomas JW. The concept of access: definition and relationship to consumer satisfaction. Med Care. 1981 Feb;19(2):127-40. doi: 10.1097/00005650-198102000-00001).

Reviewer #2: This is a very interesting topic and article that offers an overview of quantity, type, and topics within the rural primary care literature…but I am left feeling like I didn’t learn as much as I wanted. As mentioned in limitations, identifying the interventions and their findings (what actually worked?) will come in other papers, but it’s a very lengthy article to have such limited substance. Perhaps the narrative description of the papers could be shortened (it’s mostly already in the table) and presented in other ways (bar charts or other tools), with more space allotted for background, discussion, and some analysis of content around which interventions had an impact on primary care access or outcomes. I recognize this would add a lot of work, but otherwise I feel the paper's a bit too superficial.

Some specific comments:

Introduction - it's quite brief, and should be expanded upon in 2 areas in my opinion. 1) rural/remote context, e.g., the authors don’t mention vast geography, lack of public transit, data about distribution of HCP or specialists in rural Canada (or elsewhere), or why Indigenous peoples have unique issues accessing health care (with citation/support that's currently missing). 2) Could explain health system issues and how primary care is situated within the system - primary care physicians’ work settings, role, and part of multidisciplinary teams (or not).

Line 169 - stated there was an increase in number of publications per year, yet it looks like these peaked around 2017 and then went back to almost pre-peak volume - is this consistent with all publications during this time (e.g., due to the pandemic)? Either way, it’s worth noting as it’s not a steady increase as you suggest.

Line 196 - not sure why "chronic disease diabetes" is stated rather than simply diabetes?

Line 207 - "before-after studies (n=42), randomized controlled trials (n=32)." needs and between rather than comma

Line 224 - "prospective cohort studies (n=15), cohort studies (n=14), retrospective cohort studies (n=11)" - it's unclear to me what a cohort study is that is neither prospective or retrospective.

Line 353 - "rurality is associated with a 2-5 year deficit in life expectancy (2,5,392,393) which does not compare to the effect of widely studied diseases such as diabetes, heart disease and stroke.(394) However, the far greater prevalence suggests that ..." Consider rewording this to increase clarity.

Reference #1 - United Nations D of E and SA Population Division,. (edit for format)

6. PLOS authors have the option to publish the peer review history of their article (what does this mean?). If published, this will include your full peer review and any attached files.

Reviewer #1: **Yes: **Jim Ricca

Reviewer #2: No

---

## [Author Response · Author response to Decision Letter 0]

24 Mar 2024

Dear Dr. Yourkavitch

We would like to thank you and the reviewers for the thoughtful comments and suggestions on our review. In particular, we agree with your overall comment about providing additional high-level content analysis and more depth about the gaps and areas for further research. Our point-by-point response is below. 

Journal Requirements:

Thank you we have consulted and followed the style templates.

Thank you we have removed any mention of funding in the manuscript.

3. Thank you for stating the following in the Competing Interests section: "KAB and JR are family physicians who previously practiced in rural areas. KAB, JR, and SA are rural health services researchers with an interest in the equity of health service distribution. The authors have no other disclosures to report."

Please confirm that this does not alter your adherence to all PLOS ONE policies on sharing data and materials, by including the following statement: "This does not alter our adherence to PLOS ONE policies on sharing data and materials." (as detailed online in our guide for authors http://journals.plos.org/plosone/s/competing-interests). If there are restrictions on sharing of data and/or materials, please state these. Please note that we cannot proceed with consideration of your article until this information has been declared.

Thank you we have updated the competing interest statement as requested above.

4. We note that Figure 3 in your submission contain map/satellite images which may be copyrighted. All PLOS content is published under the Creative Commons Attribution License (CC BY 4.0), which means that the manuscript, images, and Supporting Information files will be freely available online, and any third party is permitted to access, download, copy, distribute, and use these materials in any way, even commercially, with proper attribution. For these reasons, we cannot publish previously copyrighted maps or satellite images created using proprietary data, such as Google software (Google Maps, Street View, and Earth). For more information, see our copyright guidelines: http://journals.plos.org/plosone/s/licenses-and-copyright.

a. You may seek permission from the original copyright holder of Figure 3 to publish the content specifically under the CC BY 4.0 license.

"I request permission for the open-access journal PLOS ONE to publish XXX under the Creative Commons Attribution License (CCAL) CC BY 4.0 (http://creativecommons.org/licenses/by/4.0/). Please be aware that this license allows unrestricted use and distribution, even commercially, by third parties. Please reply and provide explicit written permission to publish XXX under a CC BY license and complete the attached form."

In the figure caption of the copyrighted figure, please include the following text: "Reprinted from [ref] under a CC BY license, with permission from [name of publisher], original copyright [original copyright year]."

b. If you are unable to obtain permission from the original copyright holder to publish these figures under the CC BY 4.0 license or if the copyright holder's requirements are incompatible with the CC BY 4.0 license, please either i) remove the figure or ii) supply a replacement figure that complies with the CC BY 4.0 license. Please check copyright information on all replacement figures and update the figure caption with source information. If applicable, please specify in the figure caption text when a figure is similar but not identical to the original image and is therefore for illustrative purposes only.

USGS EROS (Earth Resources Observatory and Science (EROS) Center) (public domain): http://eros.usgs.gov/# Natural Earth (public domain): http://www.naturalearthdata.com/

Thank you for pointing this out. To comply with the Creative Commons Attribution License, we have removed “Figure 3: Map depicting number of studies from different countries” from the manuscript and replaced it with “Table 3 – Number of studies by country, geographic region, and economic status” (See page 11, line 359 and page 12-13, line 369-371). By making this change, we believe have been able to provide more valuable information as well so we thank you for requesting the change.

5. We note that this manuscript is a systematic review or meta-analysis; our author guidelines therefore require that you use PRISMA guidance to help improve reporting quality of this type of study. Please upload copies of the completed PRISMA checklist as Supporting Information with a file name "PRISMA checklist".

Thank you. We have uploaded an up-to-date version of the PRISMA checklist.

 

Editor Comments:

1. Thank you for submitting this interesting manuscript. In general, we believe this treatment is too superficial. We understand that more in-depth treatment will be given to topics in subsequent papers but this paper should not merely be an appetizer. Readers should learn something from your effort. Try to give at least some high-level content analysis in the body of the article and more depth about the gaps and areas for further research. Please also respond to each comment from the reviewers.

Thank you for your reflective comment on how we can add a more meaningful objective to our review that will provide the reader with more useful information about the topic areas particularly with respect for future research. While we had originally anticipated that this review would adhere to more of the basic purposes of a scoping review outlined by Joanna Briggs Institute and by Munn et al. in their paper entitled Systematic review or scoping review? Guidance for authors when choosing between a systematic or scoping review approach. BMC Med Res Methodol 18, 143 (2018), we have now included a further two objectives that include providing a high level content summary and areas for future research as you recommend and we think that this was an excellent recommendation and hopefully our response adds the extra value to the review and for the reader. The extra information has resulted in additional text throughout the manuscript (intro, methods, results and discussion) and additional tables within the results section. All additional information and tables are highlighted in track changes and/or red font.

 

Reviewer #1 - Comments to the Author

1. Introduction: The authors don't clearly and explicitly state the definition or framework they are using for Primary Care or Primary Health Care. They seem to be screening articles mainly by level of care and excluding specialist or hospital-delivered care. But are they using any kind of characteristics of primary care, like the classic 4 C's (first contact, coordination, comprehensiveness, and continuity). If the common feature of included articles is only that they are at the most basic level of care (i.e., "first contact"), but not necessarily that the care is coordinated or comprehensive, for instance, then are all included articles truly about "PHC" or "PC" or would they more accurately be described as about "basic health care" or "non-specialist care." I take as an illustrative example of what I mean by this distinction one of the interventions listed in Appendix 3 on Clinical Outcomes. From what is listed in the table, it is not clear if this was a "vertically delivered" intervention that was not integrated into other health care. Perhaps it was integrated into a coordinated, comprehensive, and continuous system of care. But I don't feel that there is enough information given for the reader to confirm that is the case.

Any interventions to improve a service that typically lives in primary care.

* Shakya, 2016, Nepal

* Uncontrolled before/after

* The primary purpose of this study was to assess the knowledge of cervical cancer among women in rural Nepal and explore the feasibility and impact of a community-based awareness program on cervical cancer. Code: Patient Education/Navigation

* Community-based educational meetings on cervical cancer and its prevention were conducted among women's groups in rural Nepal.

* Questionnaires with open-ended and closed-ended questions were administered through a face-to-face interview. All interviews were performed by the primary investigator and a doctor trained in interviewing. The questionnaire consisted of sociodemographic information (age, education level, and income source), personal information (age at marriage, number of live births, number of marriages), and questions on knowledge and attitudes related to cervical cancer to determine the outcome measures of interest. The messages that were emphasized during the education session were: (1) cervical cancer is preventable, (2) risk factors, signs, symptoms, and asymptomatic nature of early cervical cancer, and (3) the importance of undergoing gynaecological examination and cervical screening. The participants were asked whether they would participate in an upcoming cost-free cervical cancer screening program two weeks after the educational meeting.

Thank you for this comment. If we are understanding the comment correctly, the issue is that we have not clarified our definition for primary care and/or primary healthcare and our inclusion/exclusion criteria for the same. We agree our definition and criteria were not explicit and we apologize for any confusion. Our aim was to be broad in definition and we used the Canadian Foundation for Healthcare Improvements’ policy document to guide many of the definitions used in this review. As such we adhered to their definition of primary care as; “an inclusive term to cover the spectrum of first-contact healthcare models from those whose focus is comprehensive, person-centered care, sustained over time, to those that also incorporate health promotion, community development and intersectoral action to address the social determinants of health”. In piloting this definition during our screening process, we realized reporting was not always clear and we were lenient in our inclusion of studies that simply described their study as being set or delivered in primary care. We have added this information to the setting subsection in the methods. We hope this adequately addresses the comment above, if we have mis-interpreted the comment we would be happy to revisit this as needed.

2. The authors searched the following databases: PubMed, CINAHL, Cochrane Library, Embase, but did not attempt to include grey literature in their review. Can they give a reason for this?

Thank you for this question. Consultation with the medical librarian suggested that our database search was broad and had returned a fairly extensive number of peer‐reviewed articles (~7000) and as such, the risk of missing more higher quality studies was low and a grey literature search was not required to identify more records. 

3. The authors present their search terms for an illustrative database (MEDLINE). I'm curious: It does not seem that the authors included any term on cost or cost-effectiveness although this is one of the sub-outcomes under Efficiency outcomes that they list in Table 3.

Thank you for this observation. The search was developed with a medical librarian and the search did not include any terms for the outcome categories. The search was intended to be broad focusing at the level of terms for primary care or primary care type services. By doing so, we aimed not to limit by any one outcome. We acknowledge it may be possible that article indexed solely on the term of cost may have been missed but it is determined by the team that the risk of including an AND term for outcomes would make the search too specific and the likelihood of studies being solely indexed on terms related to outcome would be low. The librarian who conducted our search has taken a different position within our University and was not available to formally respond during our response time frame but may be able to respond further if needed. 

4. In the PRISMA diagram the authors ought to list the reasons for exclusion (and how frequently the reason was used to exclude an article).

Thank you, we have updated the PRISMA diagram to include this information.

5. I am bit confused by the categorization of some issues as "outcomes" and others as strategies." For instance, on Page 6, "recruitment of family physicians" and "retention of family physicians" are listed as sub-outcomes under Access, rather than as strategies or interventions that are meant to improve outcomes like availability.

Thank you for this comment. The three topics under access “recruitment of family physicians, retention of family physicians and recruitment/use of alternative providers were actually meant to be strategy topic areas under the overall outcome of access. This has been changed. 

6. Related to the last point, I'm not sure how or why the authors decided to sub-divide the outcome of efficiency. A fairly well-known sub-division of this concept is the following: availability, accessibility, accommodation, affordability, acceptability (Penchansky R, Thomas JW. The concept of access: definition and relationship to consumer satisfaction. Med Care. 1981 Feb;19(2):127-40. doi: 10.1097/00005650-198102000-00001).

Thank you for suggesting this additional reference. Our working definition for this review was based on the Canadian Foundation for Healthcare Improvements’ 2012 policy document, entitled Toward a Primary Care Strategy for Canada. In this document, efficiency in primary care is described as a primary care system that continually seeks to reduce waste and cost of supplies, equipment, space, capital, ideas, time and opportunities. We thus created the categories of cost, healthcare use and wait-times based on this description. While we can’t change our working definitions or categories now, we acknowledge that many definitions and frameworks exist so we had added a line to our limitations section to acknowledge this point.

Reviewer #2 - Comments to the Author

1. This is a very interesting topic and article that offers an overview of quantity, type, and topics within the rural primary care literature. But I am left feeling like I didn't learn as much as I wanted. As mentioned in limitations, identifying the interventions and their findings (what actually worked?) will come in other papers, but it's a very lengthy article to have such limited substance. Perhaps the narrative description of the papers could b

---

## [Decision Letter · Decision Letter 1]

17 Apr 2024

PONE-D-23-28342R1Interventions to improve primary healthcare in rural settings: A scoping reviewPLOS ONE

Dear Dr. Aubrey-Bassler,

Thank you for submitting your manuscript to PLOS ONE. After careful consideration, we feel that it has merit but does not fully meet PLOS ONE’s publication criteria as it currently stands. Therefore, we invite you to submit a revised version of the manuscript that addresses the points raised during the review process.

We look forward to receiving your revised manuscript.

Kind regards,

Jennifer Yourkavitch

Academic Editor

PLOS ONE

Journal Requirements:

Additional Editor Comments:

Thank you for the revisions. Please address the remaining few comments from the reviewer.

Reviewers' comments:

Reviewer's Responses to Questions

**Comments to the Author**

1. If the authors have adequately addressed your comments raised in a previous round of review and you feel that this manuscript is now acceptable for publication, you may indicate that here to bypass the “Comments to the Author” section, enter your conflict of interest statement in the “Confidential to Editor” section, and submit your "Accept" recommendation.

Reviewer #1: (No Response)

Reviewer #2: All comments have been addressed

2. Is the manuscript technically sound, and do the data support the conclusions?

Reviewer #1: Yes

Reviewer #2: (No Response)

3. Has the statistical analysis been performed appropriately and rigorously? 

Reviewer #1: N/A

Reviewer #2: (No Response)

4. Have the authors made all data underlying the findings in their manuscript fully available?

Reviewer #1: Yes

Reviewer #2: (No Response)

5. Is the manuscript presented in an intelligible fashion and written in standard English?

Reviewer #1: Yes

Reviewer #2: (No Response)

6. Review Comments to the Author

**Reviewer #1:** Thank you for your thoughtful responses to my comments and to those of the other review adn the editor. I find the manuscript clearer now. I especially like the clear focus on pointed out issues for further in-depth review. I find the discussion more in-depth and the findings in each sub-section of the results better summarized.

I only add that I think that one of my previous comments was not completely addressed. I am speaking of my first comment:

• The authors don’t clearly and explicitly state the definition or framework they are using for Primary Care or Primary Health Care. They seem to be screening articles mainly by level of care and excluding specialist or hospital-delivered care. But are they using any kind of characteristics of primary care, like the classic 4 C’s (first contact, coordination, comprehensiveness, and continuity). If the common feature of included articles is only that they are at the most basic level of care (i.e., “first contact”), but not necessarily that the care is coordinated or comprehensive, for instance, then are all included articles truly about “PHC” or “PC” or would they more accurately be described as about “basic health care” or “non-specialist care.”

I appreciate that you added a sentence in the limiatations. But I actually don't find a clear delineation/definiation of Primary Care/Primary Health Care to be a limitation. I think that having (and stating) a definition makes clear the scope of the Scoping Review. I think that a statement of hte definition you are using belongs in the Introduction. I think that some of what you wrote in your response to me should go there - Namely, that you are using the Canadian FOundation for Healthcare Improvement's definition and that this adheres to the common PC principles of First Contact, Comprehensiveness, Continuity, an Coordinated. But that you also "loosened" your criteria (I would be explict about HOW exactly you loosened the criteria......I think that this concern is most relevant when I see in the manuscript various entries for articles about mental health. Were these done in a setting that in which the services were deliverd by mental health specialists or were they integreated into a comprehensive primary care service delivery setting or system? I bring up mental health as an example, but I also have a question in my mind when I see the entires for cancer screening. I have seen such programs carried out as part of "community outreach" by tertiary care hospitals. I don't know if you encourntered any examples in the articles you reviewed. If so, did you exclude them or include them? I thnk you should make state such considerations and your decisions about them clearly in the introduction.

Other than this one remianing comment, I am quite satisfied with your responses and the improvements in the manuscript.

**Reviewer #2: **Detailed additions that address earlier concerns.

Typo on line 599: the increase in the yearly number of rural primary care must be...

7. PLOS authors have the option to publish the peer review history of their article (what does this mean?). If published, this will include your full peer review and any attached files.

Reviewer #1: **Yes: **Jim Ricca

Reviewer #2: No

---

## [Author Response · Author response to Decision Letter 1]

3 May 2024

Dear Dr. Yourkavitch

We would like to thank you and the reviewers for the thoughtful comments and suggestions on our review. Our point-by-point response is below:

Thank you. We have reviewed our reference list and updated some citations that were not properly ordered. No papers were retracted from our reference list.

Please include a separate legend for each figure within your manuscript file.

As requested, we have added a legend for each figure within the manuscript.

Reviewer #1: Thank you for your thoughtful responses to my comments and to those of the other review and the editor. I find the manuscript clearer now. I especially like the clear focus on pointed out issues for further in-depth review. I find the discussion more in-depth and the findings in each sub-section of the results better summarized.

I only add that I think that one of my previous comments was not completely addressed. I am speaking of my first comment:

• The authors don’t clearly and explicitly state the definition or framework they are using for Primary Care or Primary Health Care. They seem to be screening articles mainly by level of care and excluding specialist or hospital-delivered care. But are they using any kind of characteristics of primary care, like the classic 4 C’s (first contact, coordination, comprehensiveness, and continuity). If the common feature of included articles is only that they are at the most basic level of care (i.e., “first contact”), but not necessarily that the care is coordinated or comprehensive, for instance, then are all included articles truly about “PHC” or “PC” or would they more accurately be described as about “basic health care” or “non-specialist care.”

I appreciate that you added a sentence in the limitations. But I actually don't find a clear delineation/definition of Primary Care/Primary Health Care to be a limitation. I think that having (and stating) a definition makes clear the scope of the Scoping Review. I think that a statement of the definition you are using belongs in the Introduction. I think that some of what you wrote in your response to me should go there - Namely, that you are using the Canadian Foundation for Healthcare Improvement's definition and that this adheres to the common PC principles of First Contact, Comprehensiveness, Continuity, an Coordinated. But that you also "loosened" your criteria (I would be explicit about HOW exactly you loosened the criteria......I think that this concern is most relevant when I see in the manuscript various entries for articles about mental health. Were these done in a setting that in which the services were delivered by mental health specialists or were they integrated into a comprehensive primary care service delivery setting or system? I bring up mental health as an example, but I also have a question in my mind when I see the entries for cancer screening. I have seen such programs carried out as part of "community outreach" by tertiary care hospitals. I don't know if you encountered any examples in the articles you reviewed. If so, did you exclude them or include them? I think you should make state such considerations and your decisions about them clearly in the introduction.

Other than this one remaining comment, I am quite satisfied with your responses and the improvements in the manuscript.

Thank you for helping to clarify this point for us. Just to note, in our first response to this comment we did include what we wrote in our response into the methods section of the manuscript – I am not sure if that was clear from our first response. However, we also agree that we can bring this up the in the introduction as well. We didn’t want to repeat the information in both places, so we have introduced the Canadian Foundation for Healthcare Improvement's definition in the introduction and then left the information about how we used it to guide the actual inclusion/exclusion criteria and how we “loosened” the criteria in the methods section. We hope the additional information provides a clearer understanding of our selection process.

In terms of your specific comment about what was included and excluded with respect to some of the topics that might have services/interventions/programs commonly delivered in both primary care and other areas of the health system, we included the study if:

• any part of the intervention was delivered by a primary care provider individually or as part of a comprehensive or collaborative or multidisciplinary team. Note: This was the case for many of the studies that included interventions related to mental health. We excluded studies in which the intervention was solely testing an intervention delivered by a mental health specialist that had no connection to a primary care provider or setting 

OR

• it was testing a multilevel intervention and one of the interventions was delivered in primary care. For example, if the study aimed to improve referral rates/time to referral from primary care to another care service (secondary or tertiary), this was treated as an intervention to improve coordination/referral pathways of primary care. Even if the study also went on to measure patient outcomes from the secondary or tertiary care intervention.

OR

• If the intervention was initiated outside a first-contact primary care site but the aim was to improve access to primary care or improve follow-up management of conditions, etc. (this was rare but did come up once or twice in relation to screening)

OR

• If the study stated it was delivered or taking place in a primary care setting and no further details were provided

We believe the additional inclusion/exclusion criteria we added to Table 1 should cover this at a high level, however, we could include these more very specific screening rules as an appendix if that is deemed helpful.

Reviewer #2: Detailed additions that address earlier concerns.

Typo on line 599: the increase in the yearly number of rural primary care must be...

Thanks for catching this. We have fixed this typo (see tracked changes).

---

## [Decision Letter · Decision Letter 2]

3 Jun 2024

Interventions to improve primary healthcare in rural settings: A scoping review

PONE-D-23-28342R2

Dear Dr. Aubrey-Bassler,

We’re pleased to inform you that your manuscript has been judged scientifically suitable for publication and will be formally accepted for publication once it meets all outstanding technical requirements.

Kind regards,

Jennifer Yourkavitch

Academic Editor

PLOS ONE

Additional Editor Comments (optional):

Reviewers' comments:

Reviewer's Responses to Questions

**Comments to the Author**

1. If the authors have adequately addressed your comments raised in a previous round of review and you feel that this manuscript is now acceptable for publication, you may indicate that here to bypass the “Comments to the Author” section, enter your conflict of interest statement in the “Confidential to Editor” section, and submit your "Accept" recommendation.

Reviewer #1: All comments have been addressed

2. Is the manuscript technically sound, and do the data support the conclusions?

Reviewer #1: Yes

3. Has the statistical analysis been performed appropriately and rigorously? 

Reviewer #1: N/A

4. Have the authors made all data underlying the findings in their manuscript fully available?

Reviewer #1: Yes

5. Is the manuscript presented in an intelligible fashion and written in standard English?

Reviewer #1: Yes

6. Review Comments to the Author

Reviewer #1: Thank you for addressing the one remaining comment that I had. I feel that all my comments have been addressed.

7. PLOS authors have the option to publish the peer review history of their article (what does this mean?). If published, this will include your full peer review and any attached files.

Reviewer #1: **Yes: **Jim Ricca

---

## [Editor Report · Acceptance letter]

1 Jul 2024

PONE-D-23-28342R2 

PLOS ONE

Dear Dr. Aubrey-Basler, 

I'm pleased to inform you that your manuscript has been deemed suitable for publication in PLOS ONE. Congratulations! Your manuscript is now being handed over to our production team.

Kind regards, 

on behalf of

Dr. Jennifer Yourkavitch 

Academic Editor

PLOS ONE